# Cases of trisomy 21 and trisomy 18 among historic and prehistoric individuals discovered from ancient DNA

Adam Benjamin Rohrlach[1,2] ✉, Maïté Rivollat[1,3,4,5], Patxuka de-Miguel-Ibáñez [6,7,8], Ulla Nordfors [9], Anne-Mari Liira[10], João C. Teixeira [11,12,13,14], Xavier Roca-Rada [12], Javier Armendáriz-Martija[15], Kamen Boyadzhiev[16], Yavor Boyadzhiev[16], Bastien Llamas [12,13,17,18], Anthi Tiliakou[1], Angela Mötsch[1,19], Jonathan Tuke[2], Eleni-Anna Prevedorou[20], Naya Polychronakou-Sgouritsa[21], Jane Buikstra [22], Päivi Onkamo[9,23], Philipp W. Stockhammer[1,19,24], Henrike O. Heyne[25,26,27], Johannes R. Lemke [28,29], Roberto Risch [30], Stephan Schiffels [1], Johannes Krause [1], Wolfgang Haak [1] & Kay Prüfer [1] ✉

Aneuploidies, and in particular, trisomies represent the most common genetic aberrations observed in human genetics today. To explore the presence of trisomies in historic and prehistoric populations we screen nearly 10,000 ancient human individuals for the presence of three copies of any of the target autosomes. We find clear genetic evidence for six cases of trisomy 21 (Down syndrome) and one case of trisomy 18 (Edwards syndrome), and all cases are present in infant or perinatal burials. We perform comparative osteological examinations of the skeletal remains and find overlapping skeletal markers, many of which are consistent with these syndromes. Interestingly, three cases of trisomy 21, and the case of trisomy 18 were detected in two contemporaneous sites in early Iron Age Spain (800-400 BCE), potentially suggesting a higher frequency of burials of trisomy carriers in those societies. Notably, the care with which the burials were conducted, and the items found with these individuals indicate that ancient societies likely acknowledged these individuals with trisomy 18 and 21 as members of their communities, from the perspective of burial practice.

The question of how past societies were affected by and responded to disease has been a focal point of biological anthropology for decades[1,2]. The physical and mental manifestations of disease can affect the ability of individuals to function in their day-to-day life, and hence can have a direct impact on the community, shaping its response to the illness[3]. The study of pathologies in human skeletal remains from archaeological contexts has been one of the most direct approaches to understanding diseases and their effect on past communities. However, osteological examinations are limited to detecting pathologies that manifest in skeletal tissue and the causes of skeletal lesions are often not simple to diagnose, as many can be expressions of different ailments. Additionally, rare diseases remain under-represented in bioarchaeological literature due to a combination of taphonomic, methodological and public visibility factors[4,5]. Hence, a multidisciplinary approach, such as integrating genetic, anthropological and archaeological data, can provide new insights into disease, and how it might have been perceived in past societies.

Individuals with chromosomal trisomies carry three copies of a chromosome in their cells, instead of two, following an incomplete separation (disjunction) during meiosis[6]. Except in cases of trisomy mosaicism (where only some cells are affected), or partial trisomy (where only a part of a chromosome is duplicated), individuals with full autosomal trisomies rarely survive until birth. Excluding the exceedingly rare instances of live births with trisomy 22, only three types of full autosomal trisomies can be non-fatal: trisomy 21 (Down syndrome), trisomy 18 (Edwards syndrome) and trisomy 13 (Patau syndrome). However, individuals carrying trisomy 13 or trisomy 18 rarely survive into childhood without modern medical intervention[6,7].

Each case of trisomy 13, 18 and 21 is associated with congenital and sometimes severe physical and neurodevelopmental symptoms. Common symptoms for all three trisomies, such as microcephaly and brachycephaly[8], will likely also have been recognisable to past societies. Specifically, the external physical manifestations of Down syndrome usually develop with age and can lead to a missed diagnosis of the syndrome[9]. Further, it has been shown that certain regions in the genome are responsible for eight phenotypes, leading to further phenotypic variability[10]. Few documented cases of trisomies are known from history and only a handful of cases of Down syndrome have been suggested or described in anthropological reports[11–14]. Recently, a case of Down syndrome was genetically identified in Neolithic Ireland (3629–3371 BCE), although no physical description was given[15]. However, we are aware of no prehistoric or historic cases of either trisomy 13 or 18 that have been identified genetically or osteologically.

The lack of genetically identified historic and prehistoric cases of autosomal trisomies is likely due to two factors. First, the rates of prevalence of Down, Edwards and Patau syndrome today are 1:705, 1:3226 and 1:7143, respectively (when considering live births, stillborn and terminated pregnancies)[9] and, assuming similar population demographics, it can be expected that these genetic disorders occurred at the same rates in past human populations[14]. Hence, probabilistically, researchers could not have reasonably expected to find cases until ancient DNA (aDNA) data sets became large enough.

Second, methods for analysing copy number variations for modern data require long read lengths, relatively high depth of coverage and read-pair information, which are unlikely when working with aDNA[16,17]. Instead, we focus on the probabilities of reads mapping to whole chromosomes, allowing aDNA to be analysed at extremely low depths of coverage. Hence, to date, this is the first systematic genetic screening and osteological description of such cases in premodern samples.

We identify six cases of Down syndrome and one case of Edwards syndrome in a survey of 9,855 prehistoric and historic human genomes from across the globe, which are diagnosed using a novel Bayesian method designed for use with aDNA. This method can be applied to even low-coverage shotgun screened data, requiring as few as 1000 DNA sequence reads that have been mapped to the human genome. As no single osteological marker is pathognomonic for these genetic disorders, none of these cases was confirmed osteologically. However, when skeletal preservation and completeness was sufficient, we record all observed pathological lesions, and match these to osteological markers which are consistent with a diagnosis of the trisomy. By integrating evidence from archaeological contexts, we then describe the care with which many of these individuals were buried. Although it is unclear whether these individuals were identified as different by their communities of care, there is clearly no evidence that they were stigmatised by their communities in the past.

## Results
### Genetic results
We screen shotgun sequencing data from 9855 prehistoric and historic individuals (see "Methods" for descriptions of laboratory protocols, quality control measures and data processing methodology). For each individual, we record the number and proportion of reads that mapped to each of the autosomal chromosomes.

We identify six new positive cases of Down syndrome, two genetically female and four genetically male. As expected, the positive cases for Down syndrome showed an approximately 1.5-fold higher proportion of reads mapping to chromosome 21 (range: 1.44–1.52) when compared to the mean proportion for negative cases (Fig. 1). All normalised posterior probabilities are one (to machine precision) for each positive case.

We also reanalyse and confirm a previously published case of trisomy 21, the infant PN07 who was interred in a large Neolithic portal tomb in Poulnabrone, Ireland (3629–3371 BCE)[15], showing that our method can be applied to sequencing data produced in any laboratory.

Among the 9855 screened individuals, we also identify one case (CRU013) of the comparatively rarer Edwards syndrome. The proportion of reads that mapped to chromosome 18 is 1.47 times greater than for the median for the negative group (Fig. 1) and the normalised posterior probability of the individual carrying Edwards syndrome is one (to machine precision). The individual was found to be genetically female, in line with the fact that Edwards syndrome is observed in genetically female individuals more often than in genetically male individuals by a ratio of 3:2[18].

### Archaeological contexts
Of the six newly identified cases of Down syndrome, five date to between 5000 and 2400 years ago and were intramural burials (burials within settlements and not in dedicated necropoleis). The remaining case dates to the eighteenth century CE and was found within a Finnish church cemetery. We present these results in chronological order (Table 1).

The earliest case in our new data, YUN039 (2898–2700 cal BCE), is a 6-month-old, genetically female individual, discovered in a ceramic-vessel buried under the floor of a dwelling from the Early Bronze Age layer of Tell Yunatsite, from the Pazardzhik Province of southern Bulgaria. YUN039 was not found with any burial items[19,20]. LAZ019 (1398–1221 cal BCE) is a 12–16-month-old, genetically female individual,

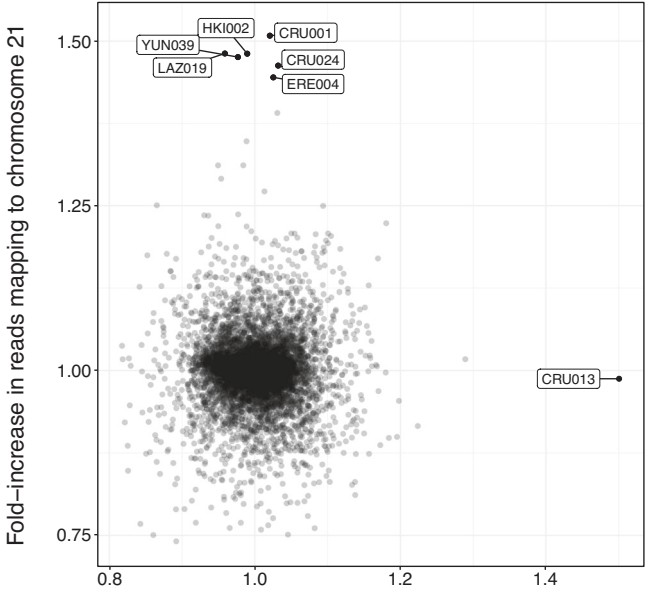

**Fig. 1 | Fold-increase in reads mapping to chromosome 18 (*x*-axis) and chromosome 21 (*y*-axis), corrected for library protocol.** Filled black circles show individuals for which there were at least 1000 reads, and for which at most one chromosome was significantly overrepresented. Labels indicate individual IDs. Source data is provided as a Source data file.

**Table 1 | Sample information for all genetically confirmed cases of trisomy 18 and 21**

| Lab/Arch ID | Region, country | Site | Date ranges | Genetic sex | Age at death | Osteological possible diagnosis | Diagnosis |
|---|---|---|---|---|---|---|---|
| CRU001/Cruz 1 | Navarra, Spain | Alto de la Cruz | 600–400 BCE[b] | XY | 38 weeks[a] | Occipital protuberance | Trisomy 21 |
| CRU013/Cruz 13 | Navarra, Spain | Alto de la Cruz | 600–400 BCE[b] | XX | 40 weeks[a] | Bone malformation (vascular increase in pars lateralis occipitalis and scapula). Unspecified nutritional deficiency | Trisomy 18 |
| CRU024/Cruz 24 | Navarra, Spain | Alto de la Cruz | 779–549 BCE (MAMS-55002) | XX | 28 weeks[a] | Nothing in particular was observed | Trisomy 21 |
| ERE004/Eretas 4 | Navarra, Spain | Las Eretas | 801–764 BCE (MAMS-55004) | XY | 26 weeks[a] | Exceptionally gracile long bones | Trisomy 21 |
| HKI002/Grave 13 | Finland | Helsinki | 1667–1800 CE[b] | XY | Full-term | Vitamin C deficiency | Trisomy 21 |
| LAZ019/LZR-25 | Aegina, Greece | Lazarides | 1398–1221 BCE (MAMS-47525) | XX | 12–16 months | Anaemia | Trisomy 21 |
| PNO7/Chamber F15 | Clare, Ireland | Poulnabrone | 3629–3371 BCE (UBA-35065) | XY | Infant | Not analysed | Trisomy 21 |
| YUN039/Burial 3 | Pazardzhik, Bulgaria | Yunatsite | 2898–2700 BCE (MAMS-45495) | XX | 6 months | Vitamin C deficiency | Trisomy 21 |

[a]Indicates the age in weeks of gestation.
[b]Indicates relative archaeological date ranges (ranges are 2-sigma calibrated when a lab number is stated).

discovered buried in a tiny cist grave in a yard-like area within the confines of a dwelling at the Mycenaean site of Lazarides on the island of Aegina, Greece[21]. LAZ019 was buried wearing a necklace made of 93 beads, glass paste, faience and carnelian (4 beads), of different colours and sizes[22].

CRU001 (600-400 BCE), CRU024 (779-549 BCE) and ERE004 (801-764 BCE) were neonatal individuals, aged ~38, ~28 and ~26 weeks gestation at the time of death, respectively[23–27]. CRU001 and ERE004 are genetically male, and CRU024 is genetically female. These remains were excavated from burials beneath dwellings found at the site of Alto de la Cruz (CRU) and Las Eretas (ERE), two of the first proto-urban centres in Early Iron Age Navarra, Spain. CRU024 was buried with rich grave goods, including bronze rings, a Mediterranean seashell, and surrounded by the complete remains of three sheep and/or goats. While CRU001 and ERE004 were apparently buried in common dwellings, CRU024 was found buried in a structure with a large, decorated fireplace, which was likely a place of ritual.

Also found at the site of Alto de la Cruz in an intramural burial was individual CRU013 (600-400 BCE), a positive case of Edwards syndrome, who was estimated to have died at ~40 weeks of gestation.

Finally, HKI002 (1640–1790 CE) was found buried in a wooden coffin under present-day Senate Square in Helsinki, previously a church graveyard in Post-Medieval Finland. The funeral attire of HKI002 contained bronze pins and decorative bronze flowers, which were a common trend of the time.

## Rates of prevalence

The six observed cases of Down syndrome in our data indicate a past rate of prevalence of ~1:1643, which is significantly lower than the overall modern rate of prevalence for Down syndrome of 1:705[9]. It is known that the age of the mother is strongly associated with an increased rate of prevalence[9]. When compared to the rate of prevalence for present-day mothers under the age of 20 (1:1282), we continue to find a lower rate of prevalence in past populations (Fig. 2), although this difference is not statistically significant. However, past rates of prevalence could be underestimated as infant burials are generally under-represented in the archaeological record[28]. Similarly, it is possible that the average age of mothers was lower due to an increased risk of death for the mother during childbirth in pre-modern times and assisted reproductive technologies allowing increased instances of motherhood at older ages in modern populations, although this lower mean age may be offset by an increased age of menarche in premodern populations[9,29–32].

The single case of Edwards syndrome indicates a premodern rate of prevalence of 1:9855, which is not statistically significantly different from the modern prevalence of 1:3226 ($p \approx 1$)[9]. Finally, observing no cases of Patau syndrome in 9855 individuals is not unexpected, given that the modern rate of prevalence is 1:7143 ($p \approx 1$)[9]. However, we caution that Edwards and Patau syndrome almost always result in pre- or perinatal deaths. Given that perinatal remains will often be poorly preserved due to reasons such as reduced bone mineralisation and smaller bone size, thus making their bones more vulnerable to destruction and their discovery more difficult, our rates of prevalence are likely underestimated[33].

## Osteological results−trisomy 21

In this section, we identify osteological markers in our sample of seven individuals that could be indications of autosomal trisomies. We report all skeletal pathologies that are either known to be related to Down or Edwards syndrome or that are observed in more than one individual. However, we caution that osteological analyses alone are insufficient for such diagnoses, as no osteological symptom is pathognomonic for trisomy 21 or 18. Moreover, a large proportion of these features could be linked to environmental factors, such as malnutrition, and to other genetic disorders that can result in abnormal growth patterns or could

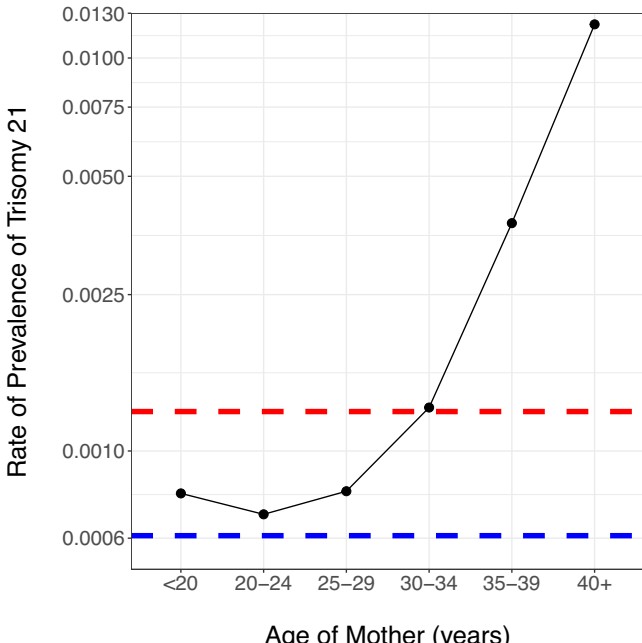

**Fig. 2 | Estimated rate of prevalence (log-scale) grouped by mother's age (from Mai et al.[9]).** The red dashed line indicates the total rate of prevalence, ignoring mother's age, and the blue dashed line indicates the observed rate of prevalence in the premodern individuals. Source data is provided as a source data file.

even fit into normal growth variability[2]. However, the genetic diagnosis of these individuals offers an opportunity to suggest a relationship of the observed features to the identified chromosomal abnormality, directly or as a consequence. The osteological study of the individuals discussed in the present work are further complicated for two reasons: first, the skeletons are incomplete and often poorly preserved (see Supplementary Methods). Second, our cases are either late fetal losses, stillbirths or, at most, 16 months of age, and not all markers may be observable[2].

To date, studies evaluating the skeletal manifestations of individuals with Down syndrome in past populations are few in number and often too statistically underpowered to answer questions about skeletal development, although studies based on modern individuals have shown that porosity and skeletal health issues, shortened femur length and reduced bone mineral density, are significantly more prevalent in cases of Down syndrome[34]. Several skeletal abnormalities have been identified in fetuses with Down syndrome, including: short stature with reduced length of long bones, microcephaly, brachycephaly, absence of nasal bone ossification and hypoplasia of the middle phalanx of the fifth digit[35]. Among the skeletal manifestations of Down syndrome that develop later in childhood are scoliosis, *pes planus* and Perthes disease[35]. Hence, we were motivated to look for abnormalities in the skeletons of the individuals that were genetically diagnosed as carrying trisomy 21.

Hand phalanges ossify at around 24 weeks of gestation, but they are so small that they are rarely recognised or correctly identified during excavation, thus making systematic recording impossible. Vitamin C, D, A, complex B and/or iron deficiency has also been associated with Down syndrome, individually or in combination, likely arising as a secondary complication of features of Down syndrome such as feeding difficulties[36]. Specifically, these deficiencies are commonly associated with cranial porosities including locations such as the sphenoid body, the external cranial vault, the superior aspect of the eye orbit (or *cribra orbitalia*), as well as on long bones, which can be anticipated in individuals with trisomy 21[37].

Despite the different stages of completeness and preservation throughout the cases, there is overlap in the anthropological observations, with the most striking being the reported growth disorders on bones, followed by porosities in the cranial elements. Overall, we observe porosity in at least one of the bones of the cranium in 4/6 cases (Supplementary Figs. 2 and 3), visible in the orbital roofs (2/3), mandible (3/4), the upper palate (2/3), the frontal and parietal bones (3/4), and/or the occipital squama (4/6). We also observe additional bone formation on the jugular limb of the *pars lateralis* in both cases for which the occipital bone was sufficiently well preserved for observation. Less common were visible *striae* or pitting of either the long bones (1/3) or the ilium (1/2). HKI002 was the only individual for which we observed (Supplementary Fig. 4) inner bone (incus) malformation (1/1), and dental enamel hypoplasia[38] (1/1) (Supplementary Fig. 5).

Other traits that can be associated with trisomy 21 overlap in three instances: individuals CRU001 and HKI002 present irregular bone growth on the *pars lateralis* segment of the occipital bone (e.g., flange, Supplementary Fig. 6), LAZ019 and YUN039 present morphological changes on the femora (Supplementary Fig. 5) (flattened and widened distal metaphysis, and antero-posterior flexion of the femur diaphysis, respectively) as well as *cribra orbitalia* and *porotic hyperostosis*. Additionally, HKI002 and YUN039 present signs of Vitamin C deficiency which often co-occurs with trisomy 21. CRU001 also displays a protrusion of the occipital squama, a cranial trait associated with trisomy 21[39] (Supplementary Fig. 7). A compilation of the traits recorded for each individual can be seen in Supplementary Table 1, and we stress that some of these observed markers, such as dental hypoplasia and bone porosity, can be caused by many other diseases and even normal growth processes.

### Osteological results—trisomy 18

Edwards syndrome is associated with very severe physical symptoms[40,41], although very few clinical studies describe skeletal malformation[42]. In total, CRU013 presents numerous bone anomalies, some not being previously reported in the literature, such as: an abnormally shaped left scapula (Supplementary Figs. 8 and 9) which could be linked to describe chest malformations[43], irregular morphology of the left hemiarch of the axis[40] (Supplementary Fig. 10), extremely thin humeral diaphysis and pronounced curvature of the epiphyseal surfaces for both humeri (Supplementary Fig. 11), thinning of the diaphysis of the right femur (left missing) and the left tibia (right missing) (Supplementary Fig. 12). Furthermore, an incomplete bone fragment, likely the diaphysis of a fibula and not a clavicula, shows an unusual twist (Supplementary Fig. 13). The age of CRU013 was estimated at 40 weeks gestation when using the maximal length of both humeri (Supplementary Table 2). Of note, however, is the underdevelopment of the distal width of the right humerus, according to which the age estimate for the same individual would have been only 30 weeks of gestation. We thus caution that estimating the age of the individual using long bones, when these bones are shown to have undergone irregular development, and when short stature is a common symptom of trisomy 18, will likely produce downward-biased estimates.

### Discussion

In this analysis of 9855 prehistoric and historic individuals, we detect six cases of Down syndrome, and one case of Edwards syndrome. In addition, we confirm one published case of Down syndrome. These individuals, all of whom died either before or shortly after birth, or at most 16 months of age, come from Neolithic Ireland (-3500 BCE), Bronze Age Bulgaria (-2700 BCE) and Greece (-1300 BCE), Iron Age Spain (-600 BCE), or Post-Medieval Finland (-1720 CE).

For the individuals who we identify as having Down syndrome, we observe evidence of porosity in the cranial bones for almost all

individuals, and abnormal growths on the *ossis occipitalis* and *pars lateralis* in both cases where these bones were sufficiently preserved. We also observe additional evidence of abnormal bone growth in occipital protrusions, inner ear bone malformation and enamel hypoplasia. Abnormal bone development, such as porosity, has been shown to occur in individuals with Down syndrome as they age, and it has also been observed in some elements in fetal development. Studies on a mouse model for Down syndrome (Ts65Dn) indicate a disruption of bone development and homoeostasis caused by trisomies of genes associated with human bone development, even as early as during the development of the fetus[44,45]. However, we stress that skeletal porosity can be caused by a number of factors, such as: taphonomic factors, many diseases of different cause, the physiological growth at different ages of the individual and the mother's health[46,47].

The skeleton associated with the case of Edwards syndrome presents with severe skeletal abnormalities, some of them being consistent with the genetic diagnosis of trisomy 18. Hyperostosis and morphological abnormalities were observed in almost all of the preserved bones. The age-at-death for this individual was estimated at 40 weeks gestation. Based on osteological criteria it is impossible to determine if this individual was stillborn or born alive.

In all reported cases, the burials of the individuals were either special, or performed with care according to standard practices. PN07 was buried in a large portal tomb in Neolithic Ireland. All of the Bronze Age or Iron Age burials were intramural burials, indicating that these infants were considered worthy of deserving a burial place inside of the dwellings. Finally, HKI002 was afforded a normal Christian burial, with burial dress following the trend of the period.

LAZ019 (12–16 months) and YUN039 (6 months) are the only cases which definitely survived postnatally. Down syndrome is associated with more than skeletal symptoms[48,49]. Affected children may have additional disorders, such as congenital heart defects, lung diseases, neurodevelopmental disorders, hearing loss, vitamin and nutrient deficiencies and possible accompanying infections. Hence, newborns with Down syndrome usually require additional care. The choice to devote more time to these individuals, suggests parents, caretakers and possibly a community with compassion that would not stigmatise or ignore those in need[50–52]. Though signs of a head injury have been recorded for individual YUN039 (i.e. extensive haemorrhages), the cause may not be interpersonal violence. The prevalence of epilepsy in Down syndrome is up to 26% and an epileptic seizure could be one of very many potential explanations for head injury in a child with Down syndrome[53]. Additionally, the haemorrhages could also be consistent with scurvy or inflammatory processes[37]. Overall, mortuary treatment provides a good indication of the attitudes communities had towards these individuals. All examples described in this study were cared for after death through various rituals, which show, in some cases exceptional, recognition of them as community members.

Although the frequency of Down syndrome cases among our tested individuals is significantly lower than the present-day rate of prevalence at birth, we caution that this difference could be explained by non-random sampling and differential sample preservation. Specifically, the remains of stillbirths, infants and children are less likely to be preserved, and are buried in prehistoric and ancient cemeteries less often than adults[28,33,54–56] Additionally, since all cases came from either stillbirths or infants, the skeletons are more fragile, less likely to be preserved and often incomplete[33]. Despite observing a lower frequency of cases of Down Syndrome, we find a total of three individuals with Down syndrome at two geographically close Iron Age sites in Spain. Additionally, the sole case of Edwards syndrome was also discovered at one of the sites. It should be noted that while the dominant burial custom in this region at the time was cremation, some perinates and infants received intramural burials. However, when considering the number of infants that have been discovered and sequenced at these specific sites thus far, the number of cases of trisomies is

surprisingly high[57]. Further research will be needed to corroborate the high frequency and to form hypotheses on the cultural practices that may have led to this.

Skeletal manifestations alone cannot be used as predictors for the presence of Down syndrome or Edwards syndrome, especially in archaeological assemblages that lack complete crania or complete long bones, such as is common in infants[33]. The manifestations of the associated traits of these syndromes are variable, and the presented cases here display levels of overlapping osteological markers, especially in the growth disorders, which feature as the primary observation in all cases. Porosities of the cranial elements follow closely in occurrence, highlighting the already established suggestion that diagnosis of such lesions be treated with caution, since they could be an expression of multiple and/or more complex pathologies.

As the aDNA record continues to grow, genetic disorders with extremely low rates of prevalence will be able to be more frequently discovered. Methods for screening data, such as the one presented here, will need to be developed and refined for better detection of these afflictionsis. Integrated with contextual and anthropological data, they afford a perspective into the way that these disorders were viewed and treated in past communities. All individuals that we identified as carrying trisomy 18 and 21 may not have been visually identifiable as "different" by their communities, given that they died in infancy or earlier. We find it notable that individuals with trisomy 18 and 21 were buried according to standard practices, or in a few cases, given exceptional burials or elaborate grave goods.

*Note added in proof*: A recent publication by Anastasiadou et al.[58] identified an infant with trisomy 21 from Iron Age Britain using ancient DNA analysis.

## Methods
### Samples
All samples were collected for their original studies with the appropriate permits as required by the laws in their associated regions. We confirm that we followed established ethical guidelines for archaeogenetic research.

### DNA sequencing and analysis
In this study, we analyse Illumina shotgun sequencing data collected from 2016 to 2022 at the Department for Archeaogenetics at the Max Planck Institute for Evolutionary Anthropology (formerly at the Max Planck Institute for the Science of Human History). Most samples are sourced from survey sequencing (used to determine sample quality for deeper downstream sequencing) of around 5–10 million reads. After adaptor trimming, reads were aligned to the human reference genome GRCh37 using the Burrows-Wheeler Aligner. Sequence reads were counted for all autosomal chromosomes after filtering for a minimum sequence length of 35 and a mapping quality of at least 25.

### Estimating posterior probabilities of cases of trisomies and calculating Z-scores
To screen the full data set for trisomy, we assume a binomial distribution for the proportion of reads that map to chromosome $c \in \{1,..,22\}$. To account for overdispersion caused by different DNA sequencing library protocols, sequencing runs and artefacts due to variable DNA preservation, we also assume a prior beta distribution for the binomial probability parameter governing the probability of mapping a read to chromosome $c$, denoted $p_c$. We estimate the parameters of the beta from a filtered subset of the data, using only samples with at least 10,000 reads (7103 samples were retained based on this criterion). Additionally, we also estimate these distributions independently for each observed type of library protocol. Note that this cut-off for parameter estimation is different to the cut-off of 1000 reads for downstream screening.

The $Z$-score for the observed proportion of reads ($\hat{p}_{i,c}$) mapping to chromosome $c$ for individual $i$, denoted $Z_{i,c}$, is calculated as the number of standard deviations of the observed proportion of reads mapping to chromosome $c$, compared to expectation via a beta-binomial distribution, as plotted in Supplementary Fig. 1. For a full derivation of the formulae, see Supplementary Methods Section 1.

### Statistical testing

We estimate the beta-binomial parameters using a maximum-likelihood approach via the *beta.mle()* function in the *Rfast* (v2.0.6) package[59]. Probability density calculations for the beta-binomial distribution are calculated using the *dbbinom()* function in the *extraDist* (v1.9.1) package[60]. Tests comparing the observed rates of prevalence to the expected modern rates of prevalence are performed using the *binom.test()* function in the *stats* package. All statistical plots are created using the *ggplot2* package[61].

### Genetic sex estimates

To estimate the genetic sex, we compare the overall fraction of reads mapping to the X and Y chromosomes for each individual, denoted $p_{i,x}$ and $p_{i,y}$, as in Roca-Rada et al.[62]. Individuals that fell within the cluster with the mean, indicating no Y chromosome, are assigned as genetically female, and the remaining individuals are assigned as genetically male.

### Osteological analyses

No reanalysis of the skeletal remains were performed. All results are collected from a reanalysis of the original publications, photographs and osteological reports. The age-at-death of infants who died close to or before birth cannot be confidently estimated until a certain stage of osteological development is reached[2]. Hence, we use multiple skeleton age estimators, based on different proposals made by researchers from forensic medicine and obstetrics[63–68]. In the case of perinatal individuals, we could only use the length of the preserved long bones to obtain an approximate age.

### Reporting summary

Further information on research design is available in the Nature Portfolio Reporting Summary linked to this article.

## Data availability

No new DNA data were generated for this study. The sequence data has been processed and we provide the proportions of reads mapping to the autosomes and total read counts as a tsv at (github.com/Ben-Rohrlach/TrisomyAncientDNAStudy[69]). However, sample names and sequencing protocols for non-reported individuals have been anonymised. The read count, and source data for Figs. 1 and 2 and Supplementary Fig. 1 can be found as Supplementary Data and at github.com/BenRohrlach/TrisomyAncientDNAStudy[69]. The current location of the remains that are analysed in this study (CRU001, CRU013, CRU024, ERE004, HKI002, LAZ019 and YUN039) are described in the Supplementary Methods, and the DNA sequences reported in this paper have been deposited in the European Nucleotide Archive under the accession number PRJEB71003. Source data are provided with this paper.

## Code availability

All analyses can be run using R (version 4.2.0), and the scripts can be found at github.com/BenRohrlach/TrisomyAncientDNAStudy[69].

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

## Acknowledgements

The authors wish to thank Professor Nigel Bean, Dr Elina Salmela, Dr Vincent Braunack-Mayer, Associate Professor Denice Higgins, Dr Jaana Hurnanen, Dr Olli Varkkola and Heini Hämäläinen (Helsinki City Museum) for enlightening and instructive conversations. This study was supported by the Max Planck Society and the European Research Council (ERC)

under the European Union's Horizon 2020 Research and Innovation Program Grant 771234-PALEoRIDER (to W.H.) and 851511 (to S.S.). J.C.T. is supported by the Australian Research Council (ARC) through a Discovery Early Career Researcher Award (DE210101235). The Max Planck - Harvard Research Center for the Archaeoscience of the Ancient Mediterranean (MHAAM) and 101001951-MySocialBeIng (to P.W.S.). R.R. is supported by The Catalan Institution for Research and Advanced Studies of the Generalitat de Catalunya.

## Author contributions

ABR and KP designed the study and SS, WH and JK edited the manuscript. ABR and KP collected and processed all of the data. ABR, KP, MR, PdMI, AT and RR wrote the first draft of the manuscript. ABR designed the Bayesian method, and ABR and KP performed all statistical analyses. JAM, KB, YB, PO, RR, UN, AML and PWS wrote the archaeological sections. All authors contributed to editing the final manuscript.

## Funding

## Competing interests

The authors declare no competing interests.

## Additional information

[1]Department of Archaeogenetics, Max Planck Institute for Evolutionary Anthropology, Leipzig, Germany. [2]School of Computer and Mathematical Sciences, University of Adelaide, Adelaide, SA, Australia. [3]ArcheOs lab, Department of Archaeology, Ghent University, Sint-Pietersnieuwstraat 35, 9000 Gent, Belgium. [4]Archaeo-DNA lab, Department of Archaeology, Durham University, Lower Mount Joy, South Road, Durham DH1 3LE, UK. [5]De la Préhistoire à l'Actuel, Culture, Environnement, Anthropologie - UMR 5199, Bordeaux University, Bât. B8, Allée Geoffroy Saint Hilaire, CS50023, 33615 Pessac cedex, France. [6]Department of Prehistory, Archaeology, Ancient History and Greek and Latin Philology, INAPH, University of Alicante, San Vicente del Raspeig, Spain. [7]Sociedad de Ciencias Aranzadi, Donosti, Spain. [8]Hospital Verge dels Lliris, Alcoi, Alicante, Spain. [9]Department of Biology, University of Turku, Turku, Finland. [10]Department of Archaeology, University of Turku, Turku, Finland. [11]Evolution of Cultural Diversity Initiative, Australian National University, Canberra, ACT, Australia. [12]Australian Centre for Ancient DNA, School of Biological Sciences, University of Adelaide, Adelaide, SA, Australia. [13]Centre of Excellence for Australian Biodiversity and Heritage, University of Adelaide, Adelaide, SA, Australia. [14]CEIS.20 Centro de Estudos Interdisciplinares, Universidade de Coimbra, Coimbra, Portugal. [15]Departamento de Ciencias Humanas y de la Educación, Universidad Pública de Navarra, Pamplona, Spain. [16]National Archaeological Institute with Museum at the Bulgarian Academy of Sciences, Saborna str. 2, Sofia, Bulgaria. [17]National Centre for Indigenous Genomics, Australian National University, Canberra, ACT, Australia. [18]Telethon Kids Institute, Indigenous Genomics Research Group, Adelaide, SA, Australia. [19]Max Planck-Harvard Research Center for the Archaeoscience of the Ancient Mediterranean (MHAAM), Max Planck Institute for Evolutionary Anthropology, Deutscher Platz 6, Leipzig, Germany. [20]Hellenic Center for Bioarchaeology, Athens, Greece. [21]Department of History and Archaeology, National and Kapodistrian University of Athens, Athens, Greece. [22]Department of Anthropology, Arizona State University, Tempe, AZ, USA. [23]Department of Biosciences, University of Helsinki, Helsinki, Finland. [24]Institute for Pre- and Protohistoric Archaeology and Archaeology of the Roman Provinces, Ludwig Maximilian University, Geschwister-Scholl-Platz 1, München, Germany. [25]Hasso-Plattner-Institute, University of Potsdam, Potsdam, Germany. [26]Hasso Plattner Institute, Mount Sinai School of Medicine, New York, USA. [27]Finnish Institute for Molecular Medicine (FIMM), University of Helsinki, Helsinki, Finland. [28]Institute of Human Genetics, University of Leipzig Medical Center, Leipzig, Germany. [29]Center for Rare Diseases, University of Leipzig Medical Center, Leipzig, Germany. [30]Departament de Prehistòria, Universitat Autònoma de Barcelona, Bellaterra, Spain. ✉e-mail: adam_ben_rohrlach@eva.mpg.de; pruefer@eva.mpg.de

