## [Peer Review File · Nature Communications]

REVIEWER COMMENTS

Reviewer #1 (Remarks to the Author):

This manuscript is well written and well informed, although I detect an inconsistency of logic on one point, which requires reframing. The authors point out how unlikely it would be that that these young trisomy individuals would be recognized as different, especially if they die prior to birth. Yet there is a hint that the authors believe it significant that these trisomy individuals were considered "members of the community." Really dealing with this issue would require an in-depth consideration of the burial patterns for other juveniles, but that really wouldn't settle the matter, if it is unlikely that the conditions would have been recognized by community members. This issue requires resolution.

Reviewer #2 (Remarks to the Author):

The paper is hugely important for palaeopathology and bioarchaeology more widely. The results are incredibly significant, particularly with regards to the presentation and diagnosis of these conditions in human remains. This is where my expertise lies, and my comments reflect this. Dividing the paper in two: for the genomic work, the analysis, interpretation and conclusions are well-presented, and to the best of my knowledge, robust and meeting accepted standards. The palaeopathology aspect of the research is very good, but it is evident that these data are heavily curtailed by the limits of the paper (i.e. word count) and by the choice of journal – the fact that an osteology journal has not been chosen- but this is entirely understandable! At present, many of the statements (e.g. line 275) are very broad-brush and could be read as showing a lack of knowledge. Please be assured that that view was not taken in the review, and the suggestions about the palaeopathology are to mitigate that possibility going-forward. It is really important that the link is made between the changes observed and the genomic findings – without that, many of the changes described could be considered by many to reflect a metabolic condition or even within 'normal' growth.

Overall, it is recommended that some extra detail is provided in the supplementary material and that information provided in the supplementary material is consistent between case studies – YUN039 is given an age estimation, whereas LAZ019 is not.

Comments (by line number)

- 1: include 'Europe' in the title, otherwise it suggests that it was a global study
- 65: it might not be possible, but could the dates be given. Note that date range/period information does not appear until line 193. Could it please be featured earlier in Introduction, even if it is just 'XX BCE to XX CE'; or refer to Table 1 and other Supplementary material
- 66: 'infant state' – change to 'infancy'; note, that it is not clear whether 'infant' is being used here to describe the biological or social age of the individual. For example, in the Roman world, individuals were considered to be infants until the age of 3 years old
- 72: 'paleoanthropology' – this term is used to describe the study of human evolution. A better fit with the paper would be 'palaeopathology', 'biological anthropology' or 'bioarchaeology'
- 90: insert 'modern' after 'without'
- 94: it is worth making the point that these would have been evident at birth, rather than later in the person's life. This is an important point, because the case-studies presented show that some individuals survived, suggesting that their care-givers and community chose not to expose them or practice infanticide
- 98: dates of the Irish Neolithic needed – or refer to Table 1
- 108-112: please briefly explain why the situation has now changed – otherwise it does read as if these issues should affect your study. The following paragraph doesn't really sufficiently explain it for readers not familiar with ancient DNA or archaeological human remains
- 163: provide citation to support the first sentence
- 165: some of these references are out-of-date, and have been revised by more recent data, such as the work of Baker, B.J., Dupras, T.L. and Tocheri, M.W., 2005. The osteology of infants and children (Vol. 12). Texas A&M University Press.

Cunningham, C., Scheuer, L. and Black, S., 2016. Developmental juvenile osteology. Academic press.

Schaefer, M., Black, S.M., Schaefer, M.C. and Scheuer, L., 2009. Juvenile osteology. London: Academic Press.

- 166: please explain why only long-bones were used to estimate age-at-death. It is possible using other skeletal elements
- 167-8: it is not clear why knowing the length of an individual was useful to the study
- 183: date of the Neolithic in Portugal or for that particular tomb – or refer to Table 1
- 200: 'age' should be capitalised
- 213: 'surrounded by three complete sheep and goats' change to 'surrounded by the remains of three complete sheep and goats'; note, you may want to delete 'complete' as non-archaeologists might not realise why you've provided that information, and perhaps change 'and' to 'an/or' as many readers might not know that without genomic analysis it is often impossible to differentiate between ovid and caprid skeletons
- 221: 'the church' – do you mean 'a church' here?
- 222: 'contained' – were the pins and flowers inside the coffin as separate items, or did they decorate the coffin or clothing (suggested by attire)? It's not clear here but line 341 suggests that it was decoration on their funeral clothes
- 234: supporting citation needed for that statement
- 235-6: 'average age of mothers': be mindful that menarch started much later in prehistoric, Roman and Medieval times, and the ages at which women married also varied considerably over the date-range of your study
- 236: 'pre-modern-medicine times' – clunky wording – 'medicine' could be deleted
- 247: please explain why these remains may be poorly preserved, or at least provide a citation
- 254-257: supporting citations needed
- 261: 'underpowered' – odd choice of words, do you mean that they did not use genomics?
- 263: be clear that 'osteoporosis and skeletal health issues' has been proven by clinical not palaeopathological data. As you used 'past populations' in lines 260-1, it does read as if these are palaeopathological findings
- 269-70: note – in the supplementary material, you do not say how the skeletons of these individuals were recovered – block lifted an excavated in a lab? Excavated in the field? Was sieving used on-site? What was the soil pH at the site? How skilled were the excavators? All of those factors also determine bone survival and recovery
- 273-5: citations needed here. Be mindful that individuals born to mothers with deficiencies also have skeletal changes at birth – see Brickley, M.B., Ives, R. and Mays, S., 2020. The bioarchaeology of metabolic bone disease. Academic Press
- 275: differential diagnoses – please use Brickley et al. (2020)
- 277-88: it is appreciated that this is explored in more detail in the supplementary material, but here it is crucial that citations are given – especially for the bone-within-bone formation. You also need to be clear about how you distinguished these changes from those associated with 'normal' growth – when/where does 'normal range' end and 'pathological' begin?
- 284: which method/criteria was used to diagnose the enamel hypoplastic defect? The image looks more like a hypermineralisation rather than a hypoplastic defect – citing the method would clarify this
- 287: what was the method of diagnosis here, and information needs to be provided about the radiographic analysis (kV, was it digital etc...)
- 290-295: many of the changes described here are seen in this age-group who have been born with vitamin D deficiency or who suffered from both scurvy and rickets. Readers need to know how and why it was determined that these changes are indicative/suggestive of trisomy 21 rather than these individuals having trisomy 21 as well as scurvy-rickets, rickets or scurvy. Citing Brickley et al. (2020) will help here, but such information is required in the supplementary material – the table isn't enough
- 300-309: the only citation provided is for changes to the axial skeleton, when the majority of listed changes concern the appendicular skeleton. A PubMed search shows that there is very limited literature, and even more so for infants with this syndrome. However, the observed changes and the clinical literature need to be brought-together to show a relationship between the syndrome and the atypical bone morphology. If these are not reported in the clinical literature,

then that also needs to be mentioned

- 301: 'scapula' – were the changes only observed in the left scapula (as per S8)? Its not clear in the manuscript or supplementary information. Or should this read 'scapulae'?
- 301-2: changes to the cervical vertebra – changes similar in appearance have been reported in the literature as being caused by aneurysms or soft-tissue pressure – just a suggestion!

Antoine, D. and Waldron, T., 2023. 10 Abnormalities of the Vertebral Artery. *The Bioarchaeology of Cardiovascular Disease*, 91, p.174.

Vaswani HA & Waldron, M., 1997. The earliest case of extracranial aneurysm of the vertebral artery. *British journal of neurosurgery*, 11(2), pp.164-165.

Waldron, T. and Antoine, D., 2002. Tortuosity or aneurysm? The palaeopathology of some abnormalities of the vertebral artery. *International Journal of Osteoarchaeology*, 12(2), pp.79-88.

- 314: 'the Neolithic in Ireland' can be change to 'Neolithic Ireland'
- 327-9: provide citations
- 328-9: it is not clear why skeletal porosity would be caused by the community's treatment of the mother – do you mean withholding food from her? Poor care? Please make this clearer
- 331-2: see comments for lines 300-309
- 334-5: it might be possible to resolve this through micro CT scanning to examine the bone taphonomy. See the work of Booth, T.J., 2020. Using bone histology to identify stillborn infants in the archaeological record. *The mother-infant nexus in anthropology: Small beginnings, significant outcomes*, pp.193-209.
- 337-41: these are important points, particularly as you rightly observe that fewer subadults are recovered from prehistoric contexts compared to adults. However, the majority burial rites must be summarised somewhere for each site – the information in lines 366-8 is sufficient for that one case-study in the manuscript, but it's not enough to carry all the results and case-studies. This needs to be provided in order for the importance of this information to be recognised by a reader
- 343: please give their estimated ages-at-death to support this point
- 343-44: supporting citations needed
- 347: change 'attention' to 'care'
- 347-8: draw on the work of Tilley here- it can just be cited to support these points

<https://openresearch-repository.anu.edu.au/handle/1885/156409>

Tilley, L. and Oxenham, M.F., 2011. Survival against the odds: Modeling the social implications of care provision to seriously disabled individuals. *International Journal of Paleopathology*, 1(1), pp.35-42.

Tilley, L., 2017. Showing that they cared: An introduction to thinking, theory and practice in the bioarchaeology of care. *New Developments in the Bioarchaeology of Care: Further Case Studies and Expanded Theory*, pp.11-43.

- 361-2: supporting citations needed
- 362-3: supporting citations needed – note, it is not proven that these are stillbirths, they may have been premature births
- 370-71: it might be worth noting here that paleogenomic research is challenging existing interpretations of lineages and biological relationships – e.g. Fowler, C., Olalde, I., Cummings, V., Armit, I., Büster, L., Cuthbert, S., Rohland, N., Cheronet, O., Pinhasi, R. and Reich, D., 2022. A high-resolution picture of kinship practices in an Early Neolithic tomb. *Nature*, 601(7894), pp.584-587.
- 375: supporting citation needed for this statement, see comments for lines 269-70
- 373-80: supporting citations needed; this finding needs to be expanded in the supplementary material
- 387-8: thank you for making this point, as it is really important that these findings are reported with empathy

S4

Please ensure that the same information is given for each site – for example, image and access to the remains is provided for Yunatsite but isn't for Lazarides.

Yunatsite

- Are urn-burials typical of subadult burials in the Bronze Age?
- What method was used to determine sex?
- Do you mean humerii, radii here? So 'paired humeral, radial' etc... Please check plural spellings
- What is Bols I1 Tibia?
- What was the bone preservation like?
- How was the individual excavated?
- Supporting citations are needed

Lazarides

- What was the bone preservation like?
- How was the individual excavated?
- Are subadults often included in chamber tomb burials?
- What is the age-at-death for this individual?
- Supporting citations are needed

Alto de la Cruz

- What was the bone preservation like?
- How was the individual excavated?
- Supporting citations are needed
- 'Youngest occupation phase' change to 'earliest occupation phase'
- Remind a reader that genomics was used to establish the sex of the individual with Down syndrome
- 'Child' – is this their biological or social age?
- CRU024: are these foot or hand phalanges, or was it not possible to establish that?

Helsinki Senate Square

- What was the bone preservation like?
- How was the individual excavated?
- Supporting citations are needed
- HKI002 is inconsistently described in the supplementary material- as a newborn, as 0 months (replace with fullterm or 40-42 weeks- if the data show this)
- "were not only considered a problem" – should 'only' be deleted here?

Tables

- S1: change 'osteological markers' to 'osteological observations' – 'markers' should only be used if these changes are reported in the clinical data and the table is showing how and where the case-studies diverge or meet these data
- S1: a statement about how these changes were scored as pathological rather than normal variation needs adding here – see comments for 277-95
- S2: is the 'hmm' a typo? Do you mean mm?
- S2: change 'measure' to 'measurement'

Figures

- Copyright information for the images needs to be given
- S2: the view(s) need to be given for each image, e.g. antero-superior
- S3: view needs to be given, and which side are these bones from?
- S4: view needs to be given (i.e. labial); note comment for line 284
- S5: view needs to be given; see note for line 287
- S6: view needs to be given
- S7: for clarity, it would be worth having 'normal' occipital pictured as well, because unless you know what you're looking at, it would be difficult to see why this one was unusual. Is a lateral view available? As when contrasted with a normal occipital, this might make the change clearer
- S8: as above, having images of the right scapula (if non-pathological) would help show the changes, as even with the red outline, it would not be clear to someone without osteological knowledge

- S9: scales are missing from several of the images; please explain which views are shown
- S10: the caption seems to be incomplete here – the image seems to also include normal humeri as a comparator (really helpful, thank you!)
- S11: views of the bones and drawings need to be explained

Reviewer #3 (Remarks to the Author):

This paper is a very interesting and important contribution to the history of rare diseases, specifically trisomies. For the first time, several cases of trisomy 21 could be detected reliably using genetic testing and the first case of trisomy 18 was found in prehistoric and historic individuals. As the osteological diagnosis of trisomy 21 is particularly challenging because the disorder is a complex of symptoms with very variable expression in the skeleton, so far only three possible osteological diagnosed cases are known. This paper shows the feasibility to find cases by screening a large number of genetic samples which otherwise would have remained unrecognized. The interdisciplinary approach including osteological changes of the detected skeletons and the socio-archaeological background are very much appreciated and add an important layer to the discussion of rare diseases in the past.

Some editing is necessary regarding information of the sample composition, the standardization of osteological data, and the interpretation of the cases from the archaeological context, see below:

Title

I would suggest to use trisomy 21 and 18 in the title and throughout the paper instead of the syndrome names. These can be included in the keywords.

Line 39: Finland is missing

Abstract

Line 58/59: could you please give numbers of the historic and prehistoric cases each? Please, specify how many adults and non-adults were included in the study. This information is absolutely necessary for the discussion on prevalence.

Line 63: the skeletal markers you mention later are very unspecific and not consistent with the syndromes.

Introduction

Please add the latest literature of rare diseases as this topic has received special attention within the last years as an important part of paleopathological research: There is a whole special issue in the International Journal of Paleopathology addressing the topic of rare diseases in paleopathology:

Gresky, J., Petiti, E. (Eds.) Ancient Rare Diseases: Definition and concept of rare diseases in Paleopathology, 2021, Special Issue in International Journal of Paleopathology.

<https://www.sciencedirect.com/journal/international-journal-of-paleopathology/special-issue/10FXN63DF33>

A digital atlas on ancient rare diseases (DAARD) was started in 2022 mapping the occurrence of ancient rare diseases: <https://daard.dainst.org>

A chapter about rare diseases in paleopathology was published in 2022 in a paleopathology textbook:

Gresky, J., 3.4 Fehlbildungen und seltene Krankheiten, in: Weber, J., Wahl, J., Zink, A. (Eds.), Osteologische Paläopathologie. Ein bebildertes Lehrbuch mit medizinischen Anmerkungen, Lehmanns 2022, 461-488.

Line 96: it is worth mentioning that the phenotypic expression varies a lot in trisomy. It is rather a complex of symptoms which shape the physical appearance with certain regions in the genome being responsible for eight phenotypes of trisomy 21 (Korbel et al. 2009).

Korbel JO et al. (2009) The genetic architecture of Down syndrome phenotypes revealed by high-resolution analysis of human segmental trisomies. PNAS 106: 1203112036.

Line 97: only three reliable cases have been described macroscopically in anthropological reports (your references 7,9,10). The case of reference 8 was checked by genetic analyses and the diagnosis of trisomy 21 could not be confirmed (Halle et al. 2019).

Halle U, Hähn C, Krause S, Krause-Kyora- B, Nothnagel M, Drichel D, Wahl J (2019) Die

Unsichtbaren. Menschen mit Trisomie 21 in Archäologie und Anthropologie. Bd. 42: Archäologische Informationen.

Line 105: it is difficult to assume similar population dynamics, e.g., it might be that woman did not reach older age, implying that the frequency of trisomy was less high.

Line 118: yes, there is not a single osteological marker, but a variety of symptoms can be found in individuals with trisomy, see a compilation in Rivollat et al. 2014 and Gresky 2022, page 486. The one individual with trisomy 18 is described as macroscopically different and this could have been diagnosed as disease even without genetic analysis.

Methods

Line 132: do not abbreviate BWA

Line 158: full stop is missing after et al

Line 161: age estimation in the very young infants is a challenge, especially when the teeth are not present. Are there any teeth in the seven individuals present? If so, dental age should be mentioned, or if not, it should also be stated. For these individuals all possible measurements of the skull and postcranium have to be given. To base an age estimation only on the long bones is rather difficult in individuals with diseases which have short stature as one of the main symptoms. In trisomy short stature already occurs prenatal. If long bone length is the only way to estimate age, please mention this limitation and discuss it critically.

Line 167: as your age estimation depends on bone length you cannot estimate both. If teeth are present, a length for the individual can be given and be related to age.

Results

Line 171: please give the numbers for historic and prehistoric individuals (adults and non-adults)

Line 179/188: normalise instead of normalize

Line 182: why did you not analyze the case from Rivollat et al. 2014?

Line 199/202: months instead of month

Line 228: where does 1,643 come from? It should be 9,855 or at least 2,752 when subtracted the 7,103 from line 141? It is quite difficult to calculate prevalence in a mixed sample as it is not comparable to prehistoric population structures. Particularly when it is not clear of what your sample is consisting in terms of age. Checking predominantly individuals who died in adult age would of course minimize the number of probable cases because individuals with trisomy mainly did not reach an older age.

Line 233/234: there are many reasons why comparing prevalence from prehistory/history is difficult and should be treated with caution: different burial patterns, different preservation of bones and possibly different preservation of aDNA as well?

Fig 2: prevalence of what, please add.

Line 262: osteoporosis and skeletal health issues is too vague, please specify.

Line 265/266: please, prepare a table of possible skeletal changes in trisomy 21 (according to e.g., Rivollat et al. 2014, Gresky 2022) in fetuses and state if these changes are present, absent or cannot be evaluated for all individuals.

Line 270: to improve systematic recording of skeletal changes it is necessary to add schemes of the preservation of the whole skeleton of each individual. This is inevitable for understanding of the methods used in the paper, e.g. if age estimation could only be done by long bone measures because the teeth of the individual are not preserved. In the supplementary, the present bones are stated but this is often too vague, e.g. fragment of parietal bone or tibia needs to be specified and a visual overview is best for that. For making data comparable, like in the DAARD, this information should be given in all papers about human remains.

Line 273: in the very young age group of this sample, feeding difficulties might not play a role

Line 278: arrested growth as linear enamel hypoplasia, short stature? Please, specify.

Line 279: porosity of the cranial bones in fetuses and young infants is so unspecific that I would not mention it as a possible sign for trisomy 21. Porosity in this very young age can resemble physiological growth process, and pathological variations of it are often difficult to prove.

Line 282: what kind of irregular bone growth? Needs to be specified.

Line 287: the term bone-within-bone to my knowledge is associated to osteopetrosis, intoxications with heavy metals etc. and looks different in radiograph than the changes on the picture. I would not use this term for these changes.

Line 290 +294: if you start with specific traits associated with trisomy 21 you should not end with unspecific cribra orbitalia etc. Have rather two sentences than putting all in one.

Line 294: is hyperostosis another term for porosity or is the bone really thickened? (see also line 332).

Line 299: please, add a table with the usual skeletal changes in trisomy 18 and state if the changes are present, absent or not available in this skeleton. A complete skeletal scheme of the individual with the bones and teeth present has to be added.

Line 299-309: it would be helpful to indicate which of the skeletal changes are to be expected in trisomy 18 and which are additional.

Line 306/308: again, estimating age by long bone length in an individual who has a short stature due to the underlying disease at least has to be discussed critically.

Discussion

Line 319: osteoporosis is different from porosity, it rather describes a systemic condition and does not match the changes visible on the pictures.

Line 322: osteoporosis in general should be confirmed by radiological analyses and I think that this is the wrong term here.

Line 324: could you give a reference for osteoporosis in fetal development from a clinical study?

Line 326: references are underlined

Line 328: age at death of the individual: be more precise and rather say physiological growth in different ages. Community treatment is very general, please, explain what you mean with this, access to diet?

Line 332: the lesions are distinct but "in line" might be a little too optimistic as you have stated that without genetics the diagnosis of trisomy could not have been made.

Line 350: haemorrhages at the external or internal lamina of the cranium? It does not have to be associated with injury, it can also be due to scurvy or inflammatory processes and is frequently visible in infants.

Line 353: underlined references

Line 355: five of seven cases died around birth or even before. In these, the disease might not even have been recognized so it is not exceptional that they were treated like the others (see also line 387/388) and should be interpreted with more caution.

Line 377: please explain in the results how growth arrest was diagnosed.

References

4. Ann instead of Annu

8. no hyphen in paleopathology

Table 1

Down and Edwards syndrome without 's, anyway better trisomy 21 and 18

Pazardzhik has a space before ,

Occipitalis is missing after pars lateralis

Exceptionally gracile long bones or the whole skeleton?

Supplementary

Line 94: hyphen missing in double stranded

Line 146: what is meant by paired Bols I1? Please, give an overview of the skeletal elements marked in a skeletal scheme. This accounts for all skeletons.

Line 179: 0.75 instead of 0,75, check text for similar mistakes

Line 184/191: please be more specific when using the term infants and children as they refer to different age groups

Line 210: give the site of the vertebral arch

Line 271: different fonts, add occipitalis to pars lateralis, the column osteological markers should contain only markers and no diagnosis like indication of vitamin C/D deficiency

Line 276: what is hmm?

Line 290: sphenoid instead of spheroid

Line 291: please describe what is on the picture and why you show it. In b the hyperostosis is well visible. D should be mentioned in the captions and also some explanation about the changes visible on these pictures.

Line 294: right or left bones? The malleolus has a weird shape as well.

Line 303: as this is a radiograph you should mention it and give the parameters like name of the machine, kV, etc. Bone-within-bone is a specific term which is not used for the change visible on the picture.

Line 308: more description is needed for pictures in general. Mention the view of the bone, like here: inferior surface of the skull base (sphenoid, clivus, partes laterales occipitalis). The detailed picture is of poor quality as are several others. Please, change them.

Line 318: it should be mentioned which picture shows which site. The picture of the anterior site is of very poor quality! The red outlines are not explained. Please provide a picture with and without

red lines and give some information on the malformation. The dorsal side shows blood vessel impression? Please explain.

Line 323: reduce the number of pictures, organize them better, name the views of the bone, which site it is from and exchange the very poor ones with better ones.

Line 329: mention that the comparative humeri come from an individual of the same age etc.

Line 334: especially for this bone a scale on the pictures is important. If there is a tibia present, it would help if the two bones were together on a picture for comparison. If not, the deformed bone together with the whole skeleton could give a better impression where it belongs to.

Julia Gresky

Reviewer #4 (Remarks to the Author):

The manuscript by Rohrlach et al. screens for chromosomal aneuploidies in 10,000 ancient genomes identifying 6 cases of chromosome 21 and one case of chromosome 18 trisomy. They compare the prevalence rates of identified trisomies between ancient and modern populations and discuss it in relation to maternal age and archaeological visibility. The individuals with trisomies have been further assessed in depth for their skeletal characteristics finding evidence for arrested growth and cranial porosities. The manuscript is well written and the results clearly presented. The findings are of general interest to a wide range of disciplines from medical genetics to physical anthropology and archaeology. I could not identify any major issues with the presented work.

Minor points

p. 6, ln 184 "showing that our method can be applied to standard sequencing data." – add more clarity to explain what standard sequencing data is and how this is different from shotgun sequence data used in other analyses

p. 6, ln 199 says that YUN039 was the oldest individual (6 months old) whereas 3 rows below it says that LAS019 was 12-16 months old

p. 7, Figure 1, while chromosome 21 is shorter than 18 and this can at least partly (in combination with variance in individual coverage) explain the difference in extent of dispersion, can partial trisomy 21 (e.g. at Pelleri et al. 2016, 2022 defined segments) be effectively ruled out in case of the individuals with >1.2 fold increase of chromosome 21 reads?

p. 7 rates of prevalence, while the overall prevalence at birth of DS in 10K ancient individuals compared to modern mothers does not reach the level of statistical significance, it might be worth, e.g. as for the context of the later discussion of skeletally detectable abnormalities that develop during the childhood or later, to estimate and compare prevalence of individuals who have reached childhood. Given that the oldest individual with trisomy 21 was either 6 or 16 months old this rate difference should be significant regardless of the maternal age as well as the archaeological visibility issue with neonatal remains.

REVIEWERS' COMMENTS

Reviewer #2 (Remarks to the Author):

Thank you for the reflective and detailed response to the comments. The revision is excellent, and accommodates all the requests.

Reviewer #3 (Remarks to the Author):

Dear authors,

Thank you very much for your thorough edits to the manuscript. It is a unique study using novel methods to answer so far not addressed questions regarding the occurrence of rare disease in the past and therefore contributes to increase their visibility. The main points of discussion are sufficiently addressed and I have only some small comments left to be addressed.

You made the problems regarding the osteological analyses and documentation clear. However, where possible it would be very helpful to provide the skeletal inventories, e.g. for LAZ019 and CRU013.

General:

Be consistent in using names (including the supplementary information): Femora/femurs, anemia/anaemia

Minor edits:

Line 23: different font

Line 44: delete full stop after USA

Line 55: delete The?

Lines 66-68: We performed comparative osteological examinations of the 67 skeletal remains and found overlapping skeletal markers, many of which are consistent with these 68 syndromes.

In my opinion the term consistent is still too strong to describe the occurrence of the general skeletal markers (porous bones, enamel hypoplasia etc.) and trisomies. They can occur in many diseases, some of them might be even age-related. I would rather divide the osteological evidence in 1. general changes including the above mentioned, and 2. the more specific ones such as the bone deformations.

Lines 284-285: ...skeletal development, although studies based on modern individuals have shown that porosity and skeletal health issues

Porosity and skeletal health issues are very vague and rather confusing without a further explanation.

Lines 303-306: Overall, we observed porosity 304 in at least one of the bones of the cranium in 4/6 cases (Figure S2), visible in the orbital roofs 305 (2/3), mandible (3/4), the upper palate (2/3), the frontal and parietal bones (3/4), and/or the 306 occipital squama (4/6).

See comment about general skeletal changes above. Mentioning porosities in all of these bones is such a general observation that is impossible to link it to skeletal evidence of trisomies.

Line 317: ...and YUN039 present signs of Vitamin C deficiency which is suggestive of trisomy 21.

Please, use something more neutral like: signs of Vitamin C deficiency which often cooccur in trisomy 21 (or something like that; suggestive sounds too certain)

Lines 318-319: protrusion of the occipital squama, a cranial trait associated with trisomy 21 after 14 weeks gestation

Please, add a reference

Line 364: add ossis occipitalis to pars lateralis

Lines 353-355: However, we stress that skeletal porosity can be caused by a number of factors, such as the physiological growth at different ages of the individual and the mother's health.

I would add: taphonomic factors, many diseases of different causes

References need to be intensely reworked:

Abbreviation of journals need to be consistent, length of hyphen between page numbers, etc

Line 456: delete ;

Line 482: journal and page number missing

Line 523: delete , after Prevedorou

etc.

Line 609, Table: Eretas has a different font and size; I would change to: Exceptionally gracile long bones; Check anemia, sometimes it is written with e or ae

Supplementary information:

Lines 151-152: Fragmentary cranial remains, paired humeral, radial, fragments of paired ulnar, femoral, paired Tibia, fragments of the fibula, pelvis, scapula

It is not clear what is meant with humeral, radial. Fragments or complete bones? Maybe it is better to use fragments of humeri, radii etc.

Line 180: LAZ019 seems to be a quite well-preserved individual studied recently. Could you mention here, that the skeletal changes possibly associated with trisomy 21 could not be evidenced on this skeleton? That is an important information which should not be dismissed.

Line 231: add bone after occipital

Line 306 table: no need to write occipital squama in capitals

Figure S2: occipital not in capitals, the protrusion would be better visible in a lateral view. Please, add a view of the external surface like in figure S7.

Figure S3: add bone after parietal

Figure S4 and S5: Helsinki is not written in italics

Figure S7: should be Alto de la Cruz

Figure S10: use photograph instead of photo

Figure S11: the length of the two cases, do you mean humeri? Delete s in epiphyses

Figure S12: what refers (see Figure S12) to?

References: need intense editing

Julia Gresky

Reviewer #4 (Remarks to the Author):

The authors have address all the points I raised in my report and I do not have any further issues with the manuscript

RESPONSE TO REVIEWER COMMENTS

Dear Reviewers,

We would like to thank the reviewers for their constructive and considered reviews which have substantially improved the manuscript. We wish to comment briefly on a couple of common themes.

First, we wish we could improve the details and images in the osteological analysis. However, the 9,885 individuals come from an in-house database that is incomplete. Undaunted, in each case we contacted the archaeologists and physical anthropologists, went back to the original publications, and contacted leading archaeologists from the regions. Sadly, this was not as fruitful as hoped, and we believe that we put serious effort into making the most from the few resources that we had in reporting possible observations. For example, the Bulgarian individual from Yunatsite was excavated in the 1990s, and the remains now rest in a Russian museum and cannot be accessed. The Helsinki remains have been returned to the state, and our colleagues were only granted access to them for a very short period. Lastly, the remains from Iron age Spain come from excavations that were performed in the 1940s (CRU001), 1950s (CRU013), 1908s (CRU024) and the mid-1990s (ERE004). However, we have been able to obtain new photographs of CRU013 (trisomy 18) to address the issues of preservation.

Second, there is some disagreement in the field, as well as between even the reviewers for this manuscript, about whether people could identify individuals with Down syndrome at birth. It is clear that Edwards syndrome produces significant physical symptoms that would have been identified. Hence, we do not claim that all of these individuals would have been considered “different” by their community. However, at the least, CRU013 (Edwards syndrome) would have been identified as different, and potentially a couple of the remaining cases, with a higher likelihood for the individuals that survived to and after birth. In any event, each of these individuals were buried with care and respect, which is an insight worth discussing, even if this was (in some cases) expected because they were not identified as different. We have gone to some effort to word this observation more carefully.

Please find below, our point-by-point response to the reviewers’ comments.

With kind regards,

Adam Rohrlach, Maité Rivollat and Kay Prüfer on behalf of all co-authors

Reviewer #1 (Remarks to the Author):

This manuscript is well written and well informed, although I detect an inconsistency of logic on one point, which requires reframing. The authors point out how unlikely it would be that that these young trisomy individuals would be recognized as different, especially if they die prior to birth. Yet there is a hint that the authors believe it significant that these trisomy individuals were considered "members of the community." Really dealing with this issue would require an in-depth consideration of the burial patterns for other juveniles, but that really wouldn't settle the matter, if it is unlikely that the conditions would have been recognized by community members. This issue requires resolution.

We thank the reviewer for their kind words, and for this important comment. We wish to clarify that what the reviewer describes as inconsistent logic, was not intended, but could also be misinterpreted by other readers. Hence, we address this point in the opening comments to all reviewers and identify this as an important point to address in our revised manuscript. We have edited the text to reflect this. In particular, the abstract states now: *"Notably, the care with which these burials were performed, and the burial items found with these infants, indicate that ancient societies acknowledged infants with Down and Edwards syndromes as members of their communities."* We further elaborate in the discussion section: *"Despite the fact that all cases of Down and Edwards syndrome identified in this study died at a very young age, we find it notable that, across all time periods and geographical regions, these individuals were acknowledged as members of their communities"*.

Reviewer #2 (Remarks to the Author):

The paper is hugely important for palaeopathology and bioarchaeology more widely. The results are incredibly significant, particularly with regards to the presentation and diagnosis of these conditions in human remains. This is where my expertise lies, and my comments reflect this. Dividing the paper in two: for the genomic work, the analysis, interpretation and conclusions are well-presented, and to the best of my knowledge, robust and meeting accepted standards. The palaeopathology aspect of the research is very good, but it is evident that these data are heavily curtailed by the limits of the paper (i.e. word count) and by the choice of journal – the fact that an osteology journal has not been chosen- but this is entirely understandable! At present, many of the statements (e.g. line 275) are very broad-brush and could be read as showing a lack of knowledge. Please be assured that that view was not taken in the review, and the suggestions about the palaeopathology are to mitigate that possibility going-forward. It is really important that the link is made between the changes observed and the genomic findings – without that, many of the changes described could be considered by many to reflect a metabolic condition or even within 'normal' growth.

We thank the reviewer for their statement on the significance of this study, and their detailed comments regarding the paleopathology which have helped to strongly improve this aspect of the manuscript.

For the osteological observations that we describe, we agree that no single one is diagnostic for trisomy 18 or 21 individually, and in isolation, they must be taken with caution. Indeed, many of these observations may have even been part of normal growth variability. However, because we have the genetic diagnosis available, we felt that it was

important that they *might be* symptoms of the trisomy. We have modified the text to make this statement clearer in the introduction of the section “Osteological results”.

Overall, it is recommended that some extra detail is provided in the supplementary material and that information provided in the supplementary material is consistent between case studies – YUN039 is given an age estimation, whereas LAZ019 is not.

We have now made information between site descriptions as consistent as possible.

Comments (by line number)

- 1: include ‘Europe’ in the title, otherwise it suggests that it was a global study

The data does come from a global study. However, a statistical argument could be made that we did not expect to observe cases in some regions due to much lower sampling density. Predictably, we did not find any Trisomy cases outside of Europe. This has been made clearer in the text.

- 65: it might not be possible, but could the dates be given. Note that date range/period information does not appear until line 193. Could it please be featured earlier in Introduction, even it is just ‘XX BCE to XX CE’; or refer to Table 1 and other Supplementary material

These dates have been added at line 65.

- 66: ‘infant state’ – change to ‘infancy’; note, that it is not clear whether ‘infant’ is being used here to describe the biological or social age of the individual. For example, in the Roman world, individuals were considered to be infants until the age of 3 years old

This has been changed.

- 72: ‘paleoanthropology’ – this term is used to describe the study of human evolution. A better fit with the paper would be ‘palaeopathology’, ‘biological anthropology’ or ‘bioarchaeology’

We have changed this in the text to now read “*The question of how past societies were affected by and responded to disease has been a focal point of paleoanthropology for decades*”.

- 90: insert ‘modern’ after ‘without’

This has been added to the text.

- 94: it is worth making the point that these would have been evident at birth, rather than later in the person’s life. This is an important point, because the case-studies presented show that some individuals survived, suggesting that their care-givers and community chose not to expose them or practice infanticide

We agree with the reviewer, although we take care to point out that these physical manifestations may have been more subtle, and varied, in prenatal and newborn individuals. Similarly to our response to Reviewer 1 and as stated in our opening comment, there seems to be some disagreement on whether these trisomies would have been highly noticeable at birth. We have also removed cleft palates as at 1-2%, this might not be described as “common”. However, we give observed estimates of prevalence for microcephaly and brachycephaly.

- 98: dates of the Irish Neolithic needed – or refer to Table 1

We have added the 14C date of 3629-3371 BCE to the text.

- 108-112: please briefly explain why the situation has now changed – otherwise it does read as if these issues should affect your study. The following paragraph doesn't really sufficiently explain it for readers not familiar with ancient DNA or archaeological human remains

We have changed this section to now read “*Second, methods for analyzing copy number variations for modern data require long read lengths, relatively high depth of coverage and read-pair information, which are unlikely when working with aDNA. Instead, we focus on the probabilities of reads mapping to whole chromosomes, allowing aDNA to be analysed at extremely low depths of coverage. Hence, to date, this is the first systematic genetic screening and osteological description of such cases in premodern samples.*”

- 163: provide citation to support the first sentence

We have added a citation to Lewis 2006 to the first sentence.

- 165: some of these references are out-of-date, and have been revised by more recent data, such as the work of

Baker, B.J., Dupras, T.L. and Tocheri, M.W., 2005. The osteology of infants and children (Vol. 12). Texas A&M University Press.

Cunningham, C., Scheuer, L. and Black, S., 2016. Developmental juvenile osteology. Academic press.

Schaefer, M., Black, S.M., Schaefer, M.C. and Scheuer, L., 2009. Juvenile osteology. London: Academic Press.

This has been updated thank you.

- 166: please explain why only long-bones were used to estimate age-at-death. It is possible using other skeletal elements

We chose to perform this estimation on long bones for the Spanish samples because age estimations on long bones are more reliable/have been tested on larger samples. However, we simply report the age-at-death estimates from the remaining studies when we could not perform age estimation ourselves.

- 167-8: it is not clear why knowing the length of an individual was useful to the study

This sentence has been removed.

- 183: date of the Neolithic in Portugal or for that particular tomb – or refer to Table 1

The 14C date for each individual has been moved to the first mention of the ID of the individual in each case to remove this ambiguity.

- 200: ‘age’ should be capitalised

This typo has been corrected.

- 213: ‘surrounded by three complete sheep and goats’ change to ‘surrounded by the remains of three complete sheep and goats’; note, you may want to delete ‘complete’ as non-archaeologists might not realise why you’ve provided that information, and perhaps change ‘and’ to ‘an/or’ as many readers might not know that without genomic analysis it is often impossible to differentiate between ovid and caprid skeletons

We have changed this sentence to now read “*surrounded by the complete remains of three sheep and/or goats*”. We have chosen to keep “complete” to underline the importance of this find.

- 221: ‘the church’ – do you mean ‘a church’ here?

This has been corrected in the text.

- 222: 'contained' – were the pins and flowers inside the coffin as separate items, or did they decorate the coffin or clothing (suggested by attire)? It's not clear here but line 341 suggests that it was decoration on their funeral clothes

This has been rephrased to make it clearer that they were part of the funeral clothing. The text now reads “*The grave also included bronze pins and decorative bronze flowers. During the 17th century, it was common to dress the deceased in formal clothing. At the turn of the 17th and 18th centuries, there was a shift from the use of actual costumes to funeral attire and shrouds, which were often fastened by pins. Children's funeral costumes followed the same trend as adult costumes, but their heads could have been decorated with floral crowns¹⁴, as in the case of grave 13/HKI002.*”

- 234: supporting citation needed for that statement

This has been added. We also clarified the sentence to state that burial practices for children were often different so that they are recovered less often.

- 235-6: 'average age of mothers': be mindful that menarch started much later in prehistoric, Roman and Medieval times, and the ages at which women married also varied considerably over the date-range of your study

We thank the reviewer for this interesting observation. We include the statement “*although this lower mean age may be offset by an increased age of menarche in pre-modern populations*” to the end of the sentence, as well as two references.

- 236: 'pre-modern-medicine times' – clunky wording – 'medicine' could be deleted

We have deleted “medicine”.

- 247: please explain why these remains may be poorly preserved, or at least provide a citation

We have added the explanation “*Given that perinatal remains will often be poorly preserved due to reasons such as reduced bone mineralisation and smaller bone size, thus making their bones more vulnerable to destruction and their discovery more difficult, our rates of prevalence are likely underestimated*” as well as a citation to skeletal preservation of children's remains in the archaeological record by BM Manifold.

- 254-257: supporting citations needed

We have added a reference to Lewis, M. (2017). Paleopathology of children: Identification of pathological conditions in the human skeletal remains of non-adults. Academic press.. Additionally, we have amended the sentences to now read “*The osteological study of the individuals discussed in this study are further complicated for two reasons: first, the skeletons are incomplete and often poorly preserved (see Supp section S4). Second, our cases are either late fetal losses, stillbirths or, at most, 16 months of age, and not all markers may be observable.*”.

- 261: 'underpowered' – odd choice of words, do you mean that they did not use genomics?

This was ambiguous, and we have added “statistically underpowered” to remedy this.

- 263: be clear that 'osteoporosis and skeletal health issues' has been proven by clinical not palaeopathological data. As you used 'past populations' in lines 260-1, it does read as if these are palaeopathological findings

Thank you for pointing this out. We have amended the sentence to read “*although studies based on modern individuals have shown that porosity and skeletal health issues are significantly more prevalent in cases of Down syndrome*”.

- 269-70: note – in the supplementary material, you do not say how the skeletons of these individuals were recovered – block lifted an excavated in a lab? Excavated in the field? Was sieving used on-site? What was the soil pH at the site? How skilled were the excavators? All of those factors also determine bone survival and recovery

This is true, and details (where available) have been added to the supplementary site descriptions. Please note however that as some of these excavations took place as early as the 1940s, some of these details remain unresolved.

- 273-5: citations needed here. Be mindful that individuals born to mothers with deficiencies also have skeletal changes at birth – see Brickley, M.B., Ives, R. and Mays, S., 2020. The bioarchaeology of metabolic bone disease. Academic Press

We completely agree, and had stated in the discussion that “*However, we stress that skeletal porosity can be caused by a number of factors, such as the physiological growth at different ages of the individual and the mother’s health*”. We have now added this citation to strengthen the discussion.

- 275: differential diagnoses – please use Brickley et al. (2020)

We have added this citation here also.

- 277-88: it is appreciated that this is explored in more detail in the supplementary material, but here it is crucial that citations are given – especially for the bone-within-bone formation. You also need to be clear about how you distinguished these changes from those associated with ‘normal’ growth – when/where does ‘normal range’ end and ‘pathological’ begin?

Due to uncertainty with the copyright of the image, and the concerns of the reviewer, we have removed the mention of bone-within-bone formation and the image (S5, radiograph). Please note that we aim to state where “skeletal observations” are “out of the ordinary” (such as in the case of the long bones of CRU013). However, in many cases we simply report descriptions from previous publications (dating as early as to the 1940s) that the study’s osteologists noticed, and we then provide images (where possible) to allow the reader to make their own decision.

Additionally, these cases that are reported in this study were diagnosed via genetic screening, and we include the osteological observations as a supplementary perspective. We believe that this perspective is important and extremely interesting, but ultimately not the aim of the study (given the age of some of these excavations and the limited access we have to the remains). For example, comparing levels of porosity, such as via mercury intrusion porosimetry (Turner-Walker G, et.al, Sub-micron spongiform porosity is the major ultra-structural alteration occurring in archaeological bone) is beyond the scope of this paper, although we would love to consider more focused collaborative studies of this kind in the near future.

- 284: which method/criteria was used to diagnose the enamel hypoplastic defect? The image looks more like a hypermineralisation rather than a hypoplastic defect – citing the method would clarify this

We have discussed this with two forensic odontologists who agree that this is more likely enamel hypoplasia than hypermineralisation. A citation has been added.

- 287: what was the method of diagnosis here, and information needs to be provided about the radiographic analysis (kV, was it digital etc...)

Due to uncertainty with the copyright of the image, and the concerns of the reviewer, we have removed the mention of bone-within-bone formation and the image (S5, radiograph).

• 290-295: many of the changes described here are seen in this age-group who have been born with vitamin D deficiency or who suffered from both scurvy and rickets. Readers need to know how and why it was determined that these changes are indicative/suggestive of trisomy 21 rather than these individuals having trisomy 21 as well as scurvy-rickets, rickets or scurvy. Citing Brickley et al. (2020) will help here, but such information is required in the supplementary material – the table isn't enough

We have added an additional sentence which reads “*Additionally, HKI002 and YUN039 present signs of Vitamin C deficiency which is suggestive of trisomy 21*”, which is supported by the Bickley 2020 citation.

• 300-309: the only citation provided is for changes to the axial skeleton, when the majority of listed changes concern the appendicular skeleton. A PubMed search shows that there is very limited literature, and even more so for infants with this syndrome. However, the observed changes and the clinical literature need to be brought-together to show a relationship between the syndrome and the atypical bone morphology. If these are not reported in the clinical literature, then that also needs to be mentioned

We agree that very few clinical data are published about the skeleton morphology of infants with Edwards syndrome. We have clarified which skeletal features are described in the literature or not in both relevant sections.

• 301: ‘scapula’ – were the changes only observed in the left scapula (as per S8)? Its not clear in the manuscript or supplementary information. Or should this read ‘scapulae’?

Only the left scapula was preserved, and we have now included an additional image with a right scapula from a similar aged individual from the same site (Figure S9) with the caption “*A comparison of the (a) ventral view and the (b) lateral view of the left scapula of CRU013 (right in both) and a normal right scapula from an individual of a similar age (left in both), also found at Alto de la Cruz.*”. Additionally, we have added “left scapula” to the main text.

• 301-2: changes to the cervical vertebra – changes similar in appearance have been reported in the literature as being caused by aneurysms or soft-tissue pressure – just a suggestion!

Antoine, D. and Waldron, T., 2023. 10 Abnormalities of the Vertebral Artery. The Bioarchaeology of Cardiovascular Disease, 91, p.174.

Vaswani HA & Waldron, M., 1997. The earliest case of extracranial aneurysm of the vertebral artery. British journal of neurosurgery, 11(2), pp.164-165.

Waldron, T. and Antoine, D., 2002. Tortuosity or aneurysm? The palaeopathology of some abnormalities of the vertebral artery. International Journal of Osteoarchaeology, 12(2), pp.79-88.

We thank the reviewer for this suggestion and these references. However, these examples are cases of adults and these changes do not occur in the first cervical. We therefore think they may not be directly comparable.

• 314: ‘the Neolithic in Ireland’ can be change to ‘Neolithic Ireland’

This has been changed, and similarly for the other time periods.

- 327-9: provide citations

We have added the following citations for the mother's health (Maternal diet, behaviour and offspring skeletal health, Goodfellow et al) and the age-at-death (Bone mineralisation in ex-preterm infants aged 8 years, Bowden et al).

- 328-9: it is not clear why skeletal porosity would be caused by the community's treatment of the mother – do you mean withholding food from her? Poor care? Please make this clearer

We had meant that the mother's and the child's health could be affected by, as you state above, poorer care. However, we observe that this is unclear and speculative, and the sentence has been rephrased and now reads "*However, we stress that skeletal porosity can be caused by a number of factors, such as the physiological growth at different ages of the individual and the mother's health.*"

- 331-2: see comments for lines 300-309

We agree that very few clinical data are published about the skeleton morphology of infants with Edwards syndrome. We have clarified which skeletal features are described in the literature or not in both relevant sections.

- 334-5: it might be possible to resolve this through micro CT scanning to examine the bone taphonomy. See the work of Booth, T.J., 2020. Using bone histology to identify stillborn infants in the archaeological record. The mother-infant nexus in anthropology: Small beginnings, significant outcomes, pp.193-209.

This is an excellent suggestion, but we do not have access to the bones at the moment. Special permissions would be required, as we currently only have permissions for teeth and petrous bones. Additionally, this analysis has not been performed on the other skeletons.

- 337-41: these are important points, particularly as you rightly observe that fewer subadults are recovered from prehistoric contexts compared to adults. However, the majority burial rites must be summarised somewhere for each site – the information in lines 366-8 is sufficient for that one case-study in the manuscript, but it's not enough to carry all the results and case-studies. This needs to be provided in order for the importance of this information to be recognised by a reader
In the site descriptions we include the following statements addressing the standard burial practices of the associated time periods as follows:

Yunatsite: In Yunatsite, 24 urn infant burials were found, under house floors. Urn infant burials have been found in 4 other EBA sites in Bulgaria^{xy}.

Lazarides: Based on current evidence, at Mycenaean Lazarides the very young received differential burial treatment: perinates, neonates, and young infants were buried in shallow pits inside the settlement, whereas children, adolescents and adults were buried in nearby chamber tombs.

Iron Age Spain: The dominant burial custom in this region at the time for adults was cremation, however, some perinates and infants received intramural burials.

Helsinki: The infant was buried in the churchyard according to the standard practices of the period.

- 343: please give their estimated ages-at-death to support this point

We have added the age at death for both cases.

- 343-44: supporting citations needed

We have added citations for the physical and behavioural symptoms associated with Down syndrome.

- 347: change 'attention' to 'care'

We have made this change.

- 347-8: draw on the work of Tilley here- it can just be cited to support these points

<https://openresearch-repository.anu.edu.au/handle/1885/156409>

Tilley, L. and Oxenham, M.F., 2011. Survival against the odds: Modeling the social implications of care provision to seriously disabled individuals. *International Journal of Paleopathology*, 1(1), pp.35-42.

Tilley, L., 2017. Showing that they cared: An introduction to thinking, theory and practice in the bioarchaeology of care. *New Developments in the Bioarchaeology of Care: Further Case Studies and Expanded Theory*, pp.11-43.

We thank the reviewer for these references. We have cited these important pieces of work.

- 361-2: supporting citations needed

These have been added.

- 362-3: supporting citations needed – note, it is not proven that these are stillbirths, they may have been premature births

We have changed the sentence to start with “*Additionally, since all cases came from stillbirths or infants*” to include that there may be no stillbirths here. We also repeat the previous citation for completeness.

- 370-71: it might be worth noting here that paleogenomic research is challenging existing interpretations of lineages and biological relationships – e.g. Fowler, C., Olalde, I., Cummings, V., Armit, I., Büster, L., Cuthbert, S., Rohland, N., Cheronet, O., Pinhasi, R. and Reich, D., 2022. A high-resolution picture of kinship practices in an Early Neolithic tomb. *Nature*, 601(7894), pp.584-587.

We thank the reviewer for pointing this out, however we were specifically talking about these Spanish Iron Age sites. We have changed the sentence to now read “*However, when considering the number of infants that have been discovered and sequenced at these specific sites so far, the number of cases of trisomies is surprisingly high.*”.

- 375: supporting citation needed for this statement, see comments for lines 269-70

We have added a citation to support this statement.

- 373-80: supporting citations needed; this finding needs to be expanded in the supplementary material

We have added citations for these comments, as well as adding citations to (where possible) each row in Supplementary Table S1.

- 387-8: thank you for making this point, as it is really important that these findings are reported with empathy

We agree!

S4

Please ensure that the same information is given for each site – for example, image and access to the remains is provided for Yunatsite but isn't for Lazarides.

We have made the information for each site and individual as consistent as possible.

Yunatsite

- Are urn-burials typical of subadult burials in the Bronze Age?

Yes. In Yunatsite, 24 urn infant burials were found. K. McSweeney, K. Bacvarov, V. Nikolov, D. Andreeva, C. Bonsall. *Infant burials in Early Bronze Age Bulgaria: a bioarchaeological appraisal of funerary behavior.* – In: (Editors: Vassil Nikolov, Wolfram Schier). *Der Schwarzmeerraum vom Neolithikum bis in die Früheisenzeit (6000–600 v. Chr.)*, 2016, 391–401, Rahden/Westf.d, under house floors. **Urn infant burials have been found in 4 other EBA sites in Bulgaria.**

- What method was used to determine sex?

DNA analyses.

- Do you mean humerii, radii here? So ‘paired humeral, radial’ etc... Please check plural spellings

In the Russian text in the publication they are pluralised.

- What is Bols I1 Tibia?

We apologise. This was an error in the Russian translation.

- What was the bone preservation like?

We have added the following statement “*Unfortunately, no images exist of the skeleton, and the skeleton is no longer accessible. However, from the report from the anthropologist, the bones were apparently well-preserved*”.

- How was the individual excavated?

During regular archaeological excavations of the tell site. It was found under a building 12A from level БХI (Б marks the Early Bronze Age layer). This is as much detail as we can recover from the original reports.

- Supporting citations are needed

We have added the original publication as a citation: Телл Юнаците. Эпоха бронзы. Том II, Часть первая. Москва 2007, 195, 210-211, and well as the recent genetic publication from Penske et. al.

Lazarides

- What was the bone preservation like?

We have added a citation to the full site report.

- How was the individual excavated?

We have added a citation to the full site report.

- Are subadults often included in chamber tomb burials?

We have added the following sentence to address this: “*Based on current preliminary evidence, at Mycenaean Lazarides, the very young juveniles, i.e., perinates, neonates, and young infants, were buried in shallow pits inside the settlement. However, given the small dataset and the limited skeletal evidence from the nearby cemetery, this observation should remain tentative*”.

- What is the age-at-death for this individual?

The child was estimated to be between 12-16 month of age.

- Supporting citations are needed

We have added references to the supplementary materials.

Alto de la Cruz

- What was the bone preservation like?

We have added “Overall, bone preservation was quite good, although the number of elements which could be recovered for each skeleton varied from nearly complete (CRU024) to relatively few (CRU013)”.

- How was the individual excavated?

The excavation methods are not explained in detail in the different publications of Alto de la Cruz, but it is unlikely that sieves were used before the 1980s. However, the responsible archaeologists, especially J. Maluquer de Motes, paid attention to these burials and excavated them as carefully as possible.

- Supporting citations are needed

We have added the citations from the main text to the supplementary, and we have added the specific anthropological study for CRU-24 by Mercadal et al. 1990.

- ‘Youngest occupation phase’ change to ‘earliest occupation phase’

This has been changed as suggested.

- Remind a reader that genomics was used to establish the sex of the individual with Down syndrome

We have added the sentence “All genetic sexes were assigned using genomic sequence data”.

- ‘Child’ – is this their biological or social age?

Thanks for pointing this out. We have changed “child” to “individual” to avoid confusion.

- CRU024: are these foot or hand phalanges, or was it not possible to establish that?

Both were missing. We have changed the statement to now read “CRU024: Almost complete skeleton, missing the phalanges of the hands and feet”.

Helsinki Senate Square

- What was the bone preservation like?

We have added the statement “The preservation of bones is generally poor in Finland because of the acidic soil. The skeleton of HKI002 was not complete, likely because of the acidic soil, but the remaining bones were quite well preserved”.

- How was the individual excavated?

We have added information about this.

- Supporting citations are needed

We have added contextual citations, but we point out that this individual is as yet unpublished.

- HKI002 is inconsistently described in the supplementary material- as a newborn, as 0 months (replace with fullterm or 40-42 weeks- if the data show this)

We have changed newborn to full-term in SI, and in table 1.

- “were not only considered a problem” – should ‘only’ be deleted here?

This word has been deleted.

Tables

- S1: change 'osteological markers' to 'osteological observations' – 'markers' should only be used if these changes are reported in the clinical data and the table is showing how and where the case-studies diverge or meet these data

This has been changed.

- S1: a statement about how these changes were scored as pathological rather than normal variation needs adding here – see comments for 277-95

We understand, and discuss this in the previous comment for 277-95. We have also changed the caption in Table S1 to now read " *Reported osteological observations and potential vitamin deficiencies for individuals carrying trisomy 21, based on direct bone observations (individuals CRU013, HEL002, LAZ019) or on respective reports (CRU001, CRU021, ERE004, YUN039). *indicates markers of potential interest but with no direct association to trisomy 21*".

- S2: is the 'hmm' a typo? Do you mean mm?

Yes, this typo has been corrected.

- S2: change 'measure' to 'measurement'

This has been changed.

Figures

- Copyright information for the images needs to be given

We have added ownership for each supplementary figure. We were not able to obtain final permission for the radiographs, and have replaced the radiograph of porosity for LAZ019 with new images (with permissions) and have removed the radiograph and all mention of bone-within-bone formation from the manuscript due to reviewer concerns.

- S2: the view(s) need to be given for each image, e.g. antero-superior

We have added the views.

- S3: view needs to be given, and which side are these bones from?

These have been added.

- S4: view needs to be given (i.e. labial); note comment for line 284

This has been added.

- S5: view needs to be given; see note for line 287

These have been added.

- S6: view needs to be given

We have added this information.

- S7: for clarity, it would be worth having 'normal' occipital pictured as well, because unless you know what you're looking at, it would be difficult to see why this one was unusual. Is a lateral view available? As when contrasted with a normal occipital, this might make the change clearer

We have included a third panel from a "normal" occipital from a similarly-aged individual recovered at Alto de la Cruz that did not test positive for a trisomy. We have added to this caption "(c) the external view of the occipital squama of a similarly-aged individual recovered from Alo de la Cruz (who presented no evidence for trisomy 18 or 21)".

- S8: as above, having images of the right scapula (if non-pathological) would help show the changes, as even with the red outline, it would not be clear to someone without osteological knowledge

As noted above, we have added an image with a second scapula from a similarly-aged individual recovered at Alto de la Cruz that did not test positive for a trisomy.

• S9: scales are missing from several of the images; please explain which views are shown

With the exception of the radiographs, we have added scales to all images. We have also added views for each image.

• S10: the caption seems to be incomplete here – the image seems to also include normal humerii as a comparator (really helpful, thank you!)

We have added to this (updated) image the following caption: “*anterior views of the humeri of CRU013 (Alto de la Cruz) indicating the decrease in the width of the diaphyses and epiphyses (middle) when compared to an individual of comparable age (left and right). The length of the two cases indicates a gestational age of a term newborn. The width of the distal epiphyses and the width of the diaphysis for CRU013 indicate an approximate age of 28-30 weeks*”.

• S11: views of the bones and drawings need to be explained

We have added the caption: “*incomplete bone fragment from CRU013 (Alto de la Cruz) showing an unusual twist that neither matches a normal (a) clavicle, or (b) fibula*”.

Reviewer #3 (Remarks to the Author):

This paper is a very interesting and important contribution to the history of rare diseases, specifically trisomies. For the first time, several cases of trisomy 21 could be detected reliably using genetic testing and the first case of trisomy 18 was found in prehistoric and historic individuals. As the osteological diagnosis of trisomy 21 is particularly challenging because the disorder is a complex of symptoms with very variable expression in the skeleton, so far only three possible osteological diagnosed cases are known. This paper shows the feasibility to find cases by screening a large number of genetic samples which otherwise would have remained unrecognized. The interdisciplinary approach including osteological changes of the detected skeletons and the socio-archaeological background are very much appreciated and add an important layer to the discussion of rare diseases in the past.

We thank the reviewer for their kind words.

Some editing is necessary regarding information of the sample composition, the standardization of osteological data, and the interpretation of the cases from the archaeological context, see below:

Title

I would suggest to use trisomy 21 and 18 in the title and throughout the paper instead of the syndrome names. These can be included in the keywords.

We agree, and have changed the title accordingly.

Line 39: Finland is missing

We have added this! Thank you.

Abstract

Line 58/59: could you please give numbers of the historic and prehistoric cases each? Please, specify how many adults and non-adults were included in the study. This information is absolutely necessary for the discussion on prevalence.

We agree that this information would be useful. However, this information is not available to us as many of these samples have incomplete records or are made up of bone fragments/elements which do not allow us to infer age at death. Also, as it is a worldwide study, the distinction between prehistoric and historical periods is not relevant or consistent everywhere. We are simply unable to estimate these numbers.

Line 63: the skeletal markers you mention later are very unspecific and not consistent with the syndromes.

Given the limitations of the descriptions and images of the skeletal remains, we opted to be more general with the descriptions. We have also added that these markers have either been shown to be associated with the syndromes or are of interest. Note though that due to uncertainty with the copyright of the image, and the concerns of the reviewer, we have removed the mention of bone-within-bone formation and the image (S5, radiograph).

Introduction

Please add the latest literature of rare diseases as this topic has received special attention within the last years as an important part of paleopathological research: There is a whole special issue in the International Journal of Paleopathology addressing the topic of rare diseases in paleopathology:

Gresky, J., Petiti, E. (Eds.) Ancient Rare Diseases: Definition and concept of rare diseases in Paleopathology, 2021, Special Issue in International Journal of Paleopathology.

<https://www.sciencedirect.com/journal/international-journal-of-paleopathology/special-issue/10FXN63DF33>

A digital atlas on ancient rare diseases (DAARD) was started in 2022 mapping the occurrence of ancient rare diseases: <https://daard.dainst.org>

A chapter about rare diseases in paleopathology was published in 2022 in a paleopathology textbook:

Gresky, J., 3.4 Fehlbildungen und seltene Krankheiten, in: Weber, J., Wahl, J., Zink, A. (Eds.), Osteologische Paläopathologie. Ein bebildertes Lehrbuch mit medizinischen Anmerkungen, Lehmanns 2022, 461-488.

We have added these citations. And thank you for the link to the digital atlas, it is fascinating.

Line 96: it is worth mentioning that the phenotypic expression varies a lot in trisomy. It is rather a complex of symptoms which shape the physical appearance with certain regions in the genome being responsible for eight phenotypes of trisomy 21 (Korbel et al. 2009).

Korbel JO et al. (2009) The genetic architecture of Down syndrome phenotypes revealed by high-resolution analysis of human segmental trisomies. PNAS 106: 1203112036.

We have added this citation, and in the interest of highlighting how varied phenotypes are for individuals with trisomy 21, we have changed the original sentence to two sentences which now read: “*Specifically, the external physical manifestations of Down syndrome usually develop with age and can lead to a missed diagnosis of the syndrome. Further, it has been shown that certain regions in the genome are responsible for eight phenotypes, leading to further phenotypic variability*”.

Line 97: only three reliable cases have been described macroscopically in anthropological reports (your references 7,9,10). The case of reference 8 was checked by genetic analyses and the diagnosis of trisomy 21 could not be confirmed (Halle et al. 2019).

Halle U, Hähn C, Krause S, Krause-Kyora- B, Nothnagel M, Drichel D, Wahl J (2019) Die Unsichtbaren. Menschen mit Trisomie 21 in Archäologie und Anthropologie. Bd. 42: Archäologische Informationen.

We have clarified in the text that these cases are either described or suggested by now stating “*Few documented cases of trisomies are known from history and only a handful of cases of Down syndrome have been suggested or described in anthropological reports*”.

Line 105: it is difficult to assume similar population dynamics, e.g., it might be that woman did not reach older age, implying that the frequency of trisomy was less high.

This is true, and we now change “expected” for “possible” and the sentence now reads “*it is possible that the average age of mothers would be lower*”. Additionally, we now add the possibility that the average age of menarche may have been higher in prehistory.

Line 118: yes, there is not a single osteological marker, but a variety of symptoms can be found in individuals with trisomy, see a compilation in Rivollat et al. 2014 and Gresky 2022, page 486. The one individual with trisomy 18 is described as macroscopically different and this could have been diagnosed as disease even without genetic analysis.

We agree, and have corrected the sentence to replace “identified” with “confirmed”.

Methods

Line 132: do not abbreviate BWA

We have written BWA as “Burrows-Wheeler Aligner” in full.

Line 158: full stop is missing after et al

This has been corrected.

Line 161: age estimation in the very young infants is a challenge, especially when the teeth are not present. Are there any teeth in the seven individuals present? If so, dental age should be mentioned, or if not, it should also be stated. For these individuals all possible measurements of the skull and postcranium have to be given. To base an age estimation only on the long bones is rather difficult in individuals with diseases which have short stature as one of the main symptoms. In trisomy short stature already occurs prenatal. If long bone length is the only way to estimate age, please mention this limitation and discuss it critically.

We didn’t re-analyse the skeletons, but instead reviewed osteological reports in most cases. Some tooth fragments are preserved in three cases (ERE004, HKI002, LAZ019) but the dental age has not been estimated on the bones and it is now impossible to assess with the rare available photographs that we have.

Line 167: as your age estimation depends on bone length you cannot estimate both. If teeth are present, a length for the individual can be given and be related to age.

We have removed this sentence.

Results

Line 171: please give the numbers for historic and prehistoric individuals (adults and non-adults)
Unfortunately, for the majority of these samples, we do not reliably have this information. Unfortunately, 14C dates are not generated for most samples, and cultural assignments

are often speculative. Similarly, we do not have age estimates in the overwhelming majority of samples.

Line 179/188: normalise instead of normalize

We have changed this.

Line 182: why did you not analyze the case from Rivollat et al. 2014?

We do not have access to the remains.

Line 199/202: months instead of month

Because “6-month-old” is adjectival, it has no plural form in English, unlike in the German translation.

Line 228: where does 1,643 come from? It should be 9,855 or at least 2,752 when subtracted the 7,103 from line 141? It is quite difficult to calculate prevalence in a mixed sample as it is not comparable to prehistoric population structures. Particularly when it is not clear of what your sample is consisting in terms of age. Checking predominantly individuals who died in adult age would of course minimize the number of probable cases because individuals with trisomy mainly did not reach an older age.

The estimate comes from the calculation $(6/9855)^{-1} = 1642.5$. This “one in x” calculation comes from estimating the number of individuals in which we would expect to see exactly one case. The estimator is derived from the maximum likelihood estimator for a binomial distribution.

As mentioned earlier, we simply do not have age estimates for the majority of our negative cases. We only mean that we have a prevalence of ~1:1,643 over all time, also ignoring age-at-death. We have explored including age-at-death in a work in progress that is focused on a specific time and region, however it also includes correcting for population size, and the probability of burial and discovery, at the least. Here, we opted to report this simpler estimate, and list the limitations of its interpretation.

Line 233/234: there are many reasons why comparing prevalence from prehistory/history is difficult and should be treated with caution: different burial patterns, different preservation of bones and possibly different preservation of aDNA as well?

This is true. We feel that we cover this in the discussion, and we have added citations to support these limitations.

Fig 2: prevalence of what, please add.

We have added “of Trisomy 21” to the y-axis label.

Line 262: osteoporosis and skeletal health issues is too vague, please specify.

We specified this as “porosity, growth disorder and skeletal malformations”.

Line 265/266: please, prepare a table of possible skeletal changes in trisomy 21 (according to e.g., Rivollat et al. 2014, Gresky 2022) in fetuses and state if these changes are present, absent or cannot be evaluated for all individuals.

Establishing such a table for individuals as young as those we describe here is extremely challenging, due to their age, the variation and lack of observable features at this age and the lack of available bones and descriptions. Moreover, as no single feature listed in Rivollat 2014 is pathognomonic, alone or in combination, we feel that it would not add significantly to the study to provide such a table.

Line 270: to improve systematic recording of skeletal changes it is necessary to add schemes of the preservation of the whole skeleton of each individual. This is inevitable for understanding of

the methods used in the paper, e.g. if age estimation could only be done by long bone measures because the teeth of the individual are not preserved. In the supplementary, the present bones are stated but this is often too vague, e.g. fragment of parietal bone or tibia needs to be specified and a visual overview is best for that. For making data comparable, like in the DAARD, this information should be given in all papers about human remains.

We strongly considered adding the skeletal preservation schemes as requested, and began the process, but the available information for each skeleton was (in some cases) far from complete enough to precisely fill these out. We felt that providing an inaccurate scheme would be the opposite of its purpose. Unfortunately, as explained previously, we currently have no access to most of the skeletons, and we cannot go back to the bones to provide a complete anthropological study at this stage. We present the salient points in the osteological reports as best possible, and we provide the list of the bones as precisely as possible, but sometimes the only information that remains is “Fragmentary cranial remains, vertebrae and costae.”, which we cannot use to accurately fill out a scheme. Hence, we present these results as a discussion point, and do not attempt to overstate the findings.

Line 273: in the very young age group of this sample, feeding difficulties might not play a role

This is true, although the only cases for which we report Vitamin C deficiencies are YUN019 (12-16 months) and HKI002 (approximately at birth) and so could have played a role here.

Line 278: arrested growth as linear enamel hypoplasia, short stature? Please, specify.

We do not discuss linear enamel hypoplasia at this point, we have now added “on bones” in the text for clarity.

Line 279: porosity of the cranial bones in fetuses and young infants is so unspecific that I would not mention it as a possible sign for trisomy 21. Porosity in this very young age can resemble physiological growth process, and pathological variations of it are often difficult to prove.

Porosity is unspecific to trisomy 21 as all other skeleton features, as none is pathognomonic. In this study we made the choice to describe all the bone abnormalities that are either known to be related to Down syndrome (and Edwards syndrome) or that are observed in more than one individual.

Line 282: what kind of irregular bone growth? Needs to be specified.

We have now changed this sentence to read “*We also observed additional bone formation on the jugular limb of the pars lateralis in both cases for which the occipital bone was sufficiently well preserved for observation*”.

Line 287: the term bone-within-bone to my knowledge is associated to osteopetrosis, intoxications with heavy metals etc. and looks different in radiograph than the changes on the picture. I would not use this term for these changes.

Due to uncertainty with the copyright of the image, and the concerns of the reviewer, we have removed the mention of bone-within-bone formation and the image (S5, radiograph).

Line 290 +294: if you start with specific traits associated with trisomy 21 you should not end with unspecific cribra orbitalia etc. Have rather two sentences than putting all in one.

This is true, we have modified the sentence.

Line 294: is hyperostosis another term for porosity or is the bone really thickened? (see also line 332).

In both cases the bone is indeed thicker, porotic hyperostosis is the right term.

Line 299: please, add a table with the usual skeletal changes in trisomy 18 and state if the changes are present, absent or not available in this skeleton. A complete skeletal scheme of the individual with the bones and teeth present has to be added.

We have obtained new photographs of the remains of CRU013, the individual diagnosed with trisomy 18. It should be noted that we have only fragmented cranial remains, the one axis hemiarch, the left scapula, both humeri, the left tibia and a fragment of long bone with morphological alterations, probably a fibula. We did not recover teeth or bones from the skull. It is clear to us is that multiple skeletal defects are common, and would likely manifest in other bones, and so we reported our observations with this in mind.

The “most common structural defects” that could be found affect mostly the heart, the hands and feet, the skull and facial features and the nervous system (Roberts 2016, *Anatomy of trisomy 18*; Kjaer 1996, *Pattern of Malformations in the Axial Skeleton in Human Trisomy 18 Fetuses*; Lin 2006, *Clinical characteristics and survival of trisomy 18 in a medical center in Taipei, 1988–2004*). We agree that a comparative study would be the ideal method to analyse this individual. Unfortunately, none of our elements overlap with the few well-studied markers, rendering a comparative study via these markers largely useless.

Line 299-309: it would be helpful to indicate which of the skeletal changes are to be expected in trisomy 18 and which are additional.

Please see the above comments. Hence we can only indicate what is different when compared to a “normal” skeleton of a similar age.

Line 306/308: again, estimating age by long bone length in an individual who has a short stature due to the underlying disease at least has to be discussed critically.

This is an excellent point, which we aimed to highlight when we pointed out the two different age estimates when using the length of the long bone (40 weeks) or the diameter (30 weeks). To further address this, we have added the sentence “*However, we caution that estimating the age of the individual using long bones, when these bones are shown to have undergone irregular development, and when short stature is a common symptom of trisomy 18, will likely produce downward-biased estimates*”.

Discussion

Line 319: osteoporosis is different from porosity, it rather describes a systemic condition and does not match the changes visible on the pictures.

Indeed, we meant porosity. We have modified every occurrence in the text.

Line 322: osteoporosis in general should be confirmed by radiological analyses and I think that this is the wrong term here.

Please see above.

Line 324: could you give a reference for osteoporosis in fetal development from a clinical study?

This is no longer applicable due to the changes from above.

Line 326: references are underlined

This has been fixed.

Line 328: age at death of the individual: be more precise and rather say physiological growth in different ages. Community treatment is very general, please, explain what you mean with this, access to diet?

We have removed the mention of community, and incorporated this suggestion. The sentence now reads “*However, we stress that skeletal porosity can be caused by a number of factors, such as the physiological growth at different ages of the individual and the mother’s health.*”.

Line 332: the lesions are distinct but “in line” might be a little too optimistic as you have stated that without genetics the diagnosis of trisomy could not have been made.

We have changed this sentence to read: “*The skeleton associated with the case of Edwards syndrome presented with severe skeletal abnormalities, some of them being consistent with the genetic diagnosis of trisomy 18*”.

Line 350: haemorrhages at the external or internal lamina of the cranium? It does not have to be associated with injury, it can also be due to scurvy or inflammatory processes and is frequently visible in infants.

Thanks for this comment. We have added this sentence: “*Additionally, the haemorrhages could also be consistent with scurvy or inflammatory processes*”.

Line 353: underlined references

This has been fixed.

Line 355: five of seven cases died around birth or even before. In these, the disease might not even have been recognized so it is not exceptional that they were treated like the others (see also line 387/388) and should be interpreted with more caution.

We now address this in our opening comments and thanks to all reviewers.

Line 377: please explain in the results how growth arrest was diagnosed.

The word “arrest” is too strong here. We have replaced the two occurrences by “disorder”, as already described in the results section.

References

4. Ann instead of Annu

This has been changed.

8. no hyphen in paleopathology

This has been changed.

Table 1

Down and Edwards syndrome without ´s, anyway better trisomy 21 and 18

We have changed these to trisomy 18 and 21.

Pazardzhik has a space before ,

This has been removed.

Occipitalis is missing after pars lateralis

This has been added.

Exceptionally gracile long bones or the whole skeleton?

The rest of the skeleton could not be identified, but there were many more remains, some ribs, parts of the jaw and fragmented remains of the skull (they were found together with a second individual, both partially preserved). The most significant altered remains were the long bones, the hemivertebra, the scapula, and what we have considered the fibula.

Supplementary

Line 94: hyphen missing in double stranded

This has been added.

Line 146: what is meant by paired Bols I1? Please, give an overview of the skeletal elements marked in a skeletal scheme. This accounts for all skeletons.

We apologise. Bols I1 was an error in translation. We also direct the reviewer to our earlier response to line 299 about why we believe skeletal schemes would not be possible.

Line 179: 0.75 instead of 0,75, check text for similar mistakes

Thanks for pointing this out. We have corrected this, and one other occurrence.

Line 184/191: please be more specific when using the term infants and children as they refer to different age groups

This has been changed.

Line 210: give the site of the vertebral arch

We have added that it is the left hemiarch axis.

Line 271: different fonts, add occipitalis to pars lateralis, the column osteological markers should contain only markers and no diagnosis like indication of vitamin C/D deficiency

We have changed the caption to read “Osteological observations and potential vitamin deficiencies for individuals carrying Down syndrome.”

Line 276: what is hmm?

This has been corrected to “mm”.

Line 290: sphenoid instead of spheroid

This has been changed.

Line 291: please describe what is on the picture and why you show it. In b the hyperostosis is well visible. D should be mentioned in the captions and also some explanation about the changes visible on these pictures.

This has been added.

Line 294: right or left bones? The malleolus has a weird shape as well.

Both are left bones, and this has been added to the caption.

Line 303: as this is a radiograph you should mention it and give the parameters like name of the machine, kV, etc. Bone-within-bone is a specific term which is not used for the change visible on the picture.

Due to uncertainty with the copyright of the image, and the concerns of the reviewer, we have removed the mention of bone-within-bone formation and the image (S5, radiograph).

Line 308: more description is needed for pictures in general. Mention the view of the bone, like here: inferior surface of the skull base (sphenoid, clivus, partes laterales occipitalis). The detailed picture is of poor quality as are several others. Please, change them.

Unfortunately, we do not have better-quality pictures in almost all cases, and we cannot access the bones to take new ones. We have added descriptive details in the captions to make the visualisation clearer.

Line 318: it should be mentioned which picture shows which site. The picture of the anterior site is of very poor quality! The red outlines are not explained. Please provide a picture with and without red lines and give some information on the malformation. The dorsal side shows blood vessel impression? Please explain.

We have added the unannotated images and added “with the morphological alteration indicated by the red arrow”. Unfortunately, while we could obtain new photographs of the

left scapula, we could not get close-up images of the scapula, and are forced to use the images from the original analysis.

Line 323: reduce the number of pictures, organize them better, name the views of the bone, which site it is from and exchange the very poor ones with better ones.

We have obtained a new high-resolution photograph of this element.

Line 329: mention that the comparative humeri come from an individual of the same age etc.

This has been added, and we now say “*indicating the decrease in the width of the diaphyses and epiphyses (middle) when compared to an individual of comparable age*”.

Line 334: especially for this bone a scale on the pictures is important. If there is a tibia present, it would help if the two bones were together on a picture for comparison. If not, the deformed bone together with the whole skeleton could give a better impression where it belongs to.

We have added new photographs of this bone, including two new photographs of (a) the bone with scale indicated, and (b) the bone with the complete collection of bones from CRU013.

Julia Gresky

Reviewer #4 (Remarks to the Author):

The manuscript by Rohrlach et al. screens for chromosomal aneuploidies in 10,000 ancient genomes identifying 6 cases of chromosome 21 and one case of chromosome 18 trisomy. They compare the prevalence rates of identified trisomies between ancient and modern populations and discuss it in relation to maternal age and archaeological visibility. The individuals with trisomies have been further assessed in depth for their skeletal characteristics finding evidence for arrested growth and cranial porosities. The manuscript is well written and the results clearly presented. The findings are of general interest to a wide range of disciplines from medical genetics to physical anthropology and archaeology. I could not identify any major issues with the presented work.

We thank the reviewer for their comments.

Minor points

p. 6, ln 184 “showing that our method can be applied to standard sequencing data.” – add more clarity to explain what standard sequencing data is and how this is different from shotgun sequence data used in other analyses

We agree that this statement was ambiguous. We meant that these methods work on data from other labs as the 9,855 shotgun samples came from two in-house labs (Max Planck Institute for the Science of Human History and the Max Planck Institute for Evolutionary Anthropology). We have changed the sentence to read “*our method can be applied to sequencing data produced in any laboratory*”.

p. 6, ln 199 says that YUN039 was the oldest individual (6 months old) whereas 3 rows below it says that LAS019 was 12-16 months old

We meant to say “the earliest” sample in sampling age. We have changed oldest to *earliest* to clarify this.

p. 7, Figure 1, while chromosome 21 is shorter than 18 and this can at least partly (in combination with variance in individual coverage) explain the difference in extent of dispersion, can partial trisomy 21 (e.g. at Pelleri et al. 2016, 2022 defined segments) be effectively ruled out in case of the individuals with >1.2 fold increase of chromosome 21 reads?

This is an excellent point, and the focus of future research. We found that we could not reliably statistically identify partial trisomies with shotgun screening data, and are analysing different types of sequence data to see if the resolution can be improved. However, in several “borderline” cases, we observed that other chromosomes were also differentially mapped to, indicating that these effects are likely a result of biased DNA preservation or possibly caused by biases in DNA sequencing library preparation and represent distributional outliers.

p. 7 rates of prevalence, while the overall prevalence at birth of DS in 10K ancient individuals compared to modern mothers does not reach the level of statistical significance, it might be worth, e.g. as for the context of the later discussion of skeletally detectable abnormalities that develop during the childhood or later, to estimate and compare prevalence of individuals who have reached childhood. Given that the oldest individual with trisomy 21 was either 6 or 16 months old this rate difference should be significant regardless of the maternal age as well as the archaeological visibility issue with neonatal remains.

We agree that further information on all tested individuals would have enabled us to investigate this question in greater detail. However, as stated earlier, the information about the age of death in the database is unfortunately incomplete. Over 90% of the individuals included in our study have no age estimates associated with the record in the database. Only 17 samples are labelled as perinatal and 55 as infant (0-3 years). In addition, the sampling of individuals for our database is opportunistic and cannot be considered unbiased. For these reasons we have not attempted to test prevalence data further. However, further testing might be possible in specific time frames and regions, and we are currently working on one such extension of our study.

RESPONSE TO REVIEWER COMMENTS

Dear Reviewers,

Thank you for the comments on our research article, now titled “*Discovery of cases of Trisomy 21 and Trisomy 18 among 9,855 Historic and Prehistoric Individuals*”. We appreciate the careful and insightful reviews, and the constructive suggestions. We believe that the manuscript has been significantly improved.

Please find below, our point-by-point response to the reviewer’s comments.

With kind regards,

Adam Rohrlach, Maïté Rivollat and Kay Prüfer on behalf of all co-authors

Reviewer #2’s comments:

We’d like to thank the reviewer again for their very helpful comments and suggestions. We have addressed these in a point-by-point form, below.

Main Text:

Line 23: different font

We have fixed this issue.

Line 44: delete full stop after USA

Fixed.

Line 55: delete The?

Fixed.

Lines 66-68: We performed comparative osteological examinations of the 67 skeletal remains and found overlapping skeletal markers, many of which are consistent with these 68 syndromes. In my opinion the term consistent is still too strong to describe the occurrence of the general skeletal markers (porous bones, enamel hypoplasia etc.) and trisomies. They can occur in many diseases, some of them might be even age-related. I would rather divide the osteological evidence in 1. general changes including the above mentioned, and 2. the more specific ones such as the bone deformations.

We agree that not all skeletal markers are equally consistent or correlated with trisomy 21. However, we feel that addressing this in the abstract might complicate it, and instead address this in the main text. Specifically, we have added the sentence: “we stress that some of these observed markers, such as dental hypoplasia and bone porosity, can be caused by many other diseases and even normal growth processes”.

Lines 284-285: ...skeletal development, although studies based on modern individuals have shown that porosity and skeletal health issues ...Porosity and skeletal health issues are very vague and rather confusing without an further explanation.

We have added more detail by giving two examples of skeletal health, specifically “shortened femur length and reduced bone mineral density” as observed in LaCombe & Roper 2020. We also further motivate this discussion by adding the sentence “Hence, we

were motivated to look for abnormalities in the skeletons of the individuals that were genetically diagnosed as carrying trisomy 21”.

Lines 303-306: Overall, we observed porosity 304 in at least one of the bones of the cranium in 4/6 cases (Figure S2), visible in the orbital roofs 305 (2/3), mandible (3/4), the upper palate (2/3), the frontal and parietal bones (3/4), and/or the 306 occipital squama (4/6). See comment about general skeletal changes above. Mentioning porosities in all of these bones is such a general observation that is impossible to link it to skeletal evidence of trisomies.

We agree, and point out that we have already stated in the main text that “*However, we stress that skeletal porosity can be caused by a number of factors, such as: taphonomic factors, many diseases of different cause, the physiological growth at different ages of the individual and the mother’s health*”. However, earlier in the text where we introduce these observations, we now also state “*and we stress that some of these observed markers, such as dental hypoplasia and bone porosity, can be caused by many other diseases and even normal growth processes*”.

Line 317: ...and YUN039 present signs of Vitamin C deficiency which is suggestive of trisomy 21. Please, use something more neutral like: signs of Vitamin C deficiency which often cooccur in trisomy 21 (or something like that; suggestive sounds too certain)

We have made this change.

Lines 318-319: protrusion of the occipital squama, a cranial trait associated with trisomy 21 after 14 weeks gestation Please, add a reference

This has been added.

Line 364: add ossis occipitalis to pars lateralis

Fixed

Lines 353-355: However, we stress that skeletal porosity can be caused by a number of factors, such as the physiological growth at different ages of the individual and the mother’s health. I would add: taphonomic factors, many diseases of different causes

We have made these changes.

References need to be intensely reworked:

Abbreviation of journals need to be consistent

length of hyphen between page numbers, etc

Line 456: delete ;

Fixed

Line 482: journal and page number missing

Fixed

Line 523: delete , after Prevedorou etc.

Fixed

Line 609, Table: Eretas has a different font and size; I would change to: Exceptionally gracile long bones; Check anemia, sometimes it is written with e or ae

Fixed

Supplementary information:

Lines 151-152: Fragmentary cranial remains, paired humeral, radial, fragments of paired ulnar, femoral, paired Tibia, fragments of the fibula, pelvis, scapula. It is not clear what is meant with

humeral, radial. Fragments or complete bones? Maybe it is better to use fragments of humeri, radii etc.

We followed the reviewers suggestion.

Line 180: LAZ019 seems to be a quite well-preserved individual studied recently. Could you mention here, that the skeletal changes possibly associated with trisomy 21 could not be evidenced on this skeleton? That is an important information which should not be dismissed.

We agree with the reviewer that further analysis of the skeletal markers which LAZ019 does not indicate is warranted. Unfortunately, for this paper we are limited to the published information in the osteology report for the individual. We are hopeful that our findings further motivate interest in a more detailed study of the remains in the future.

Line 231: add bone after occipital

Fixed.

Line 306 table: no need to write occipital squama in capitals

Fixed.

Figure S2:

occipital not in capitals

Fixed.

the protrusion would be better visible in a lateral view.

Please, add a view of the external surface like in figure S7.

We apologise, but these images cannot be obtained. We were simply lucky with the new images obtained for Figure S7. However, we believe that these remains will be revisited in an upcoming study, and have passed this request along to that research team.

Figure S3: add bone after parietal

Fixed.

Figure S4 and S5: Helsinki is not written in italics

Fixed.

Figure S7: should be Alto de la Cruz

Fixed.

Figure S10: use photograph instead of photo

Fixed.

Figure S11:

the length of the two cases, do you mean humeri?

We did mean humeri and fixed the error.

Delete s in episphyses

Fixed.

Figure S12: what refers (see Figure S12) to?

References: need intense editing

We have carefully read and edited the references.